# Cullin-associated and neddylation-dissociated protein 1 (CAND1) alleviates NAFLD by reducing ubiquitinated degradation of ACAA2

Xiang Huang[1,9], Xin Liu[2,9], Xingda Li[3,9], Yang Zhang [1,9], Jianjun Gao[4], Ying Yang[1], Yuan Jiang [5], Haiyu Gao[1], Chongsong Sun[1], Lina Xuan[1], Lexin Zhao[1], Jiahui Song[1], Hairong Bao[1], Zhiwen Zhou[1], Shangxuan Li[1], Xiaofang Zhang [1], Yanjie Lu[1]✉, Xiangyu Zhong [4]✉, Baofeng Yang [1,6,7]✉ & Zhenwei Pan [1,6,8]✉

Nonalcoholic fatty liver disease (NAFLD) is the most common liver disorder with high morbidity and mortality. The current study aims to explore the role of Cullin-associated and neddylation-dissociated protein 1 (CAND1) in the development of NAFLD and the underlying mechanisms. CAND1 is reduced in the liver of NAFLD male patients and high fat diet (HFD)-fed male mice. CAND1 alleviates palmitate (PA) induced lipid accumulation in vitro. Hepatocyte-specific knockout of CAND1 exacerbates HFD-induced liver injury in HFD-fed male mice, while hepatocyte-specific knockin of CAND1 ameliorates these pathological changes. Mechanistically, deficiency of CAND1 enhances the assembly of Cullin1, F-box only protein 42 (FBXO42) and acetyl-CoA acyl-transferase 2 (ACAA2) complexes, and thus promotes the ubiquitinated degradation of ACAA2. ACAA2 overexpression abolishes the exacerbated effects of CAND1 deficiency on NAFLD. Additionally, androgen receptor binds to the −187 to −2000 promoter region of *CAND1*. Collectively, CAND1 mitigates NAFLD by inhibiting Cullin1/FBXO42 mediated ACAA2 degradation.

The incidence and prevalence of nonalcoholic fatty liver disease (NAFLD) are rapidly increasing worldwide, which are regulated by multiple factors including several hormones, genetics, nutrition, and cold exposure[1–3]. NAFLD encompasses a spectrum of diseases, including nonalcoholic fatty liver disease (NAFL) and nonalcoholic steatohepatitis (NASH), which can develop into cirrhosis or even hepatocellular carcinoma (HCC)[4]. The most common pathological features in the liver of NAFLD patients are steatosis of hepatocytes,

[1]Department of Pharmacology (National Key Laboratory of Frigid Zone Cardiovascular Disease, Key Laboratory of Cardiovascular Research. Ministry of Education), College of Pharmacy, Harbin Medical University, Harbin, Heilongjiang 150086, P. R. China. [2]The Department of Histology and Embryology, Harbin Medical University, Harbin 150086, China. [3]Department of Pharmacy at the Second Affiliated Hospital, and Department of Pharmacology at College of Pharmacy (The Key Laboratory of Cardiovascular Research, Ministry of Education), Harbin Medical University, Harbin 150086, China. [4]The Department of Hepatopancreatobility, Surgery Second Affiliated Hospital of Harbin Medical University, Harbin 150086, China. [5]Medical Research Center, Sun Yat-Sen Memorial Hospital, Sun Yat-Sen University, Guangzhou 510120, China. [6]Research Unit of Noninfectious Chronic Diseases in Frigid Zone, Chinese Academy of Medical Sciences, 2019 Research Unit 070, Harbin, Heilongjiang 150086, P. R. China. [7]State Key Laboratory, Harbin Medical University, Harbin 150086, China. [8]Key Laboratory of Cell Transplantation, The First Affiliated Hospital, Harbin Medical University, Harbin 150086, China. [9]These authors contributed equally: Xiang Huang, Xin Liu, Xingda Li, Yang Zhang. ✉e-mail: yjlu@hrbmu.edu.cn; zhongxiangyu@hrbmu.edu.cn; yangbf@ems.hrbmu.edu.cn; panzw@ems.hrbmu.edu.cn

insulin resistance, inflammation, and even fibrosis[5]. Despite increasing investment in the exploration of novel molecular mechanisms and therapeutic targets of NAFLD, there are still significant challenges and difficulties in the treatment of NAFLD due to the lack of efficient therapeutic agents.

Ubiquitination of proteins is achieved through an enzymatic cascade involving ubiquitin-activating (E1), ubiquitin-conjugating (E2), and ubiquitin-ligating (E3) enzymes[6]. Ubiquitin ligases confer specificity to the system by selective ubiquitination of target proteins that are then degraded by proteasome. This process maintains efficient turnover of proteins[7]. Cullin-RING Ligases (CRLs) represent a collection of around 250 enzyme complexes that ubiquitinate protein substrates to alter their function or mark them for proteasomal degradation[8]. In the development of NAFLD, the homeostasis of many proteins was disturbed, and intervening ubiquitin-proteasome system to fine-tune target protein expression has become a potential therapeutic strategy for NAFLD. Sorting nexin 8 (SNX8) markedly blocked hepatocyte lipid deposition in NAFLD by recruiting E3 ligase tripartite motif containing 28 (TRIM28) and promoting the ubiquitination and subsequent proteasomal degradation of fatty acid synthase (FASN)[9]. FBXW5, a member of the Cullin-RING E3 family, directly ubiquitinated apoptosis signal-regulating kinase 1 (ASK1) in hepatocytes and blocked the progression of NAFLD[10].

Acetyl-CoA Acyltransferase 2 (ACAA2) is a thiolytic enzyme for the final step of fatty acid β-oxidation, which promotes the metabolism of fatty acids. Reduced expression of ACAA2 impaired fatty acid β-oxidation and eventually exacerbated kidney fibrosis in acute kidney injury[11]. Knockout of ACAA2 homolog mitochondrial trifunctional protein (MTP) reduced fatty acid oxidation capacity in the liver, and increased hepatic steatosis, accelerating the progression of NAFLD[12].

Cullin-associated and neddylation-dissociated protein 1 (CAND1) is a critical regulator of CRLs. It can modulate the formation rate of substrate receptor complex e.g. S-phase kinase-associated protein 1 (Skp1)-Cullin1-F-box protein (SCF), and then control the degradation rate of substrate proteins to achieve the balance between protein homeostasis and biological function[13]. CAND1/2 double knockout (DKO) cells displayed defects in IκBα degradation and SCF[β-TrCP] assembly to cause IκBα protein accumulation[14]. CAND1 regulates diverse biological functions and diseases. Overexpression of CAND1 led to a significant increase of cyclin-dependent kinase (CDK) inhibitor p27 to regulate adipogenesis[15,16]. CAND1 knockout contributed to rapid proliferation and high migration of lung cancer cells[17]. CAND1 deficiency inhibited the ubiquitinated degradation of calcineurin and led to heart failure[18]. However, whether CAND1 plays a role in NAFLD, particularly in the vicious cycle of hepatic steatosis, insulin resistance, and inflammation, remains to be explored.

In the present study, we mined the GEO database of NAFLD male patients and screened for the deregulated genes associated with protein degradation. Interestingly, CAND1 is one of the key downregulated genes. We, therefore, explored the regulation of CAND1 on NAFLD and the molecular mechanisms. We discovered that CAND1 was markedly downregulated during hepatic steatosis. Hepatocyte-specific overexpression of CAND1 prevented the development of NAFLD by inhibiting the assembly of Cullin1, F-box only protein (FBXO42), and acetyl-CoA acyltransferase 2 (ACAA2) complexes and the subsequent ubiquitinated degradation of ACAA2. The findings suggest that CAND1 is a potential target for the treatment of NAFLD.

## Results
### Downregulation of CAND1 in NAFLD and its influence on lipid metabolism
To identify regulatory genes on protein degradation in NAFLD, we first analyzed the GEO database of NAFLD male patients (https://www.ncbi.nlm.nih.gov/geo/query/acc.cgi?acc=GSE126848). There were 24 up-regulated genes and 8678 down-regulated genes

overlapping among differentially expressed (DE) genes of Normal and NAFL, and Normal and NASH (Supplementary Fig. 1a). Gene Ontology (GO) analysis showed that 5 significant functional items were associated with protein ubiquitination on overlapping genes (Supplementary Fig. 1b). A large number of E3 ubiquitin ligases and their regulatory genes were altered during liver steatosis, including CRLs regulator CAND1 (Supplementary Fig. 1c). Considering the unique regulatory property of CAND1 on protein degradation, we thus focused on this gene in the current study.

We examined CAND1 expression levels in liver biopsy samples from NAFLD patients and normal donors. CAND1 mRNA and protein were decreased in the liver tissues of NAFLD patients than in normal donors (Fig. 1a, b). Consistently, the mRNA and protein levels of CAND1 were significantly reduced in the livers of mice after HFD treatment compared with controls (Fig. 1c, d). In addition, CAND1 mRNA and protein levels were decreased substantially after exposure to palmitic acid (PA) for 24 hours in AML12 and THLE-2 cells (Fig. 1e–h). These results suggest a plausible role of CAND1 in the development of NAFLD.

To investigate the effects of CAND1 on lipid metabolism, we constructed siRNAs and overexpressing plasmids of CAND1 (Supplementary Fig. 2). Knockdown of CAND1 aggravated lipid accumulation after PA induction in THLE-2, AML12, and HepG2 cells (Supplementary Fig. 2b–d, 2j–l, 2r–t). Conversely, overexpression of CAND1 mitigated lipid accumulation after PA induction in THLE-2, AML12 and HepG2 cells (Supplementary Fig. 2f–h, 2n–p, 2v–x). These data demonstrated that CAND1 promoted lipid metabolism and reduced fat accumulation in vitro.

### CAND1 deficiency exacerbates HFD-induced liver injury
To explore the role of CAND1 in the development of NAFLD in vivo, we generated CAND1 global knockout (KO) mice (Supplementary Fig. 3a, b). Due to the lethality of CAND1 homozygous mice, we used heterozygous (CAND1 KO[+/-]) mice for HFD induction. Hepatic steatosis is the earliest pathological feature of NAFLD. We observed an increased liver-to-body weight ratio (LW/BW) in CAND1 KO[+/-]-HFD than in WT-HFD mice (Supplementary Fig. 3c). Consistently, the level of triglyceride (TG) was increased in livers of CAND1 KO[+/-] mice compared with WT mice after 12 weeks of HFD treatment (Supplementary Fig. 3d). Accordingly, H&E and Oil Red O staining showed that HFD treatment resulted in more lipid accumulation in the livers of CAND1 KO[+/-] than WT mice (Supplementary Fig. 3e). Insulin resistance is a common feature of NAFLD and contributes to its development[19]. We observed that CAND1 KO[+/-]-HFD mice exhibited higher fasting glucose levels, fasting insulin levels, and homeostatic model assessment of insulin resistance (HOMA-IR) compared with WT-HFD mice (Supplementary Fig. 3f–h). Furthermore, CAND1 KO[+/-]-HFD mice developed more serve glucose intolerance and insulin resistance upon HFD challenge than WT-HFD mice, as revealed by the glucose tolerance test (GTT) and insulin tolerance test (ITT) (Supplementary Fig. 3i, j).

To verify the specific effects of CAND1 on hepatocytes, we constructed hepatocyte-specific CAND1 knockout (conditional knockout, cKO) male mice (Supplementary Fig. 4a, b). Of note, there was no statistical difference in food intake between WT and CAND1 cKO mice after HFD treatment (Fig. 2a). HFD treatment increased body weight and epididymal fat weight of WT and CAND1 cKO mice after 16 weeks of HFD, and the body weight and epididymal fat weight were comparable between WT-HFD and CAND1 cKO-HFD mice (Fig. 2b, c). The LB/BW ratio was increased in CAND1 cKO-HFD mice (Fig. 2d). The levels of TG and TC were significantly elevated in CAND1 cKO-HFD mice compared with WT-HFD mice (Fig. 2e, f). H&E and Oil Red O staining showed that lipid accumulation was more severe in the livers of CAND1 cKO-HFD than WT-HFD mice (Fig. 2g,

h). CNAD1 cKO-HFD mice displayed higher fasting blood glucose, insulin concentrations and HOMA-IR (Fig. 2i–k). Moreover, CAND1 cKO-HFD mice exhibited exacerbated glucose tolerance and insulin sensitivity compared with WT-HFD mice (Fig. 2l–o). Liver inflammation is a common pathological change in the progression of NAFL to NASH[20]. The livers of CAND1 cKO-HFD mice exhibited increased

expression of pro-inflammatory factors compared with WT-HFD mice (Fig. 2p). However, fibrotic area and serum αFP (liver cancer marker) level had no difference between WT-HFD and CAND1 cKO-HFD mice (Supplementary Fig. 6a, c). These findings indicate that CAND1 deficiency in hepatocytes aggravates HFD-induced liver injury.

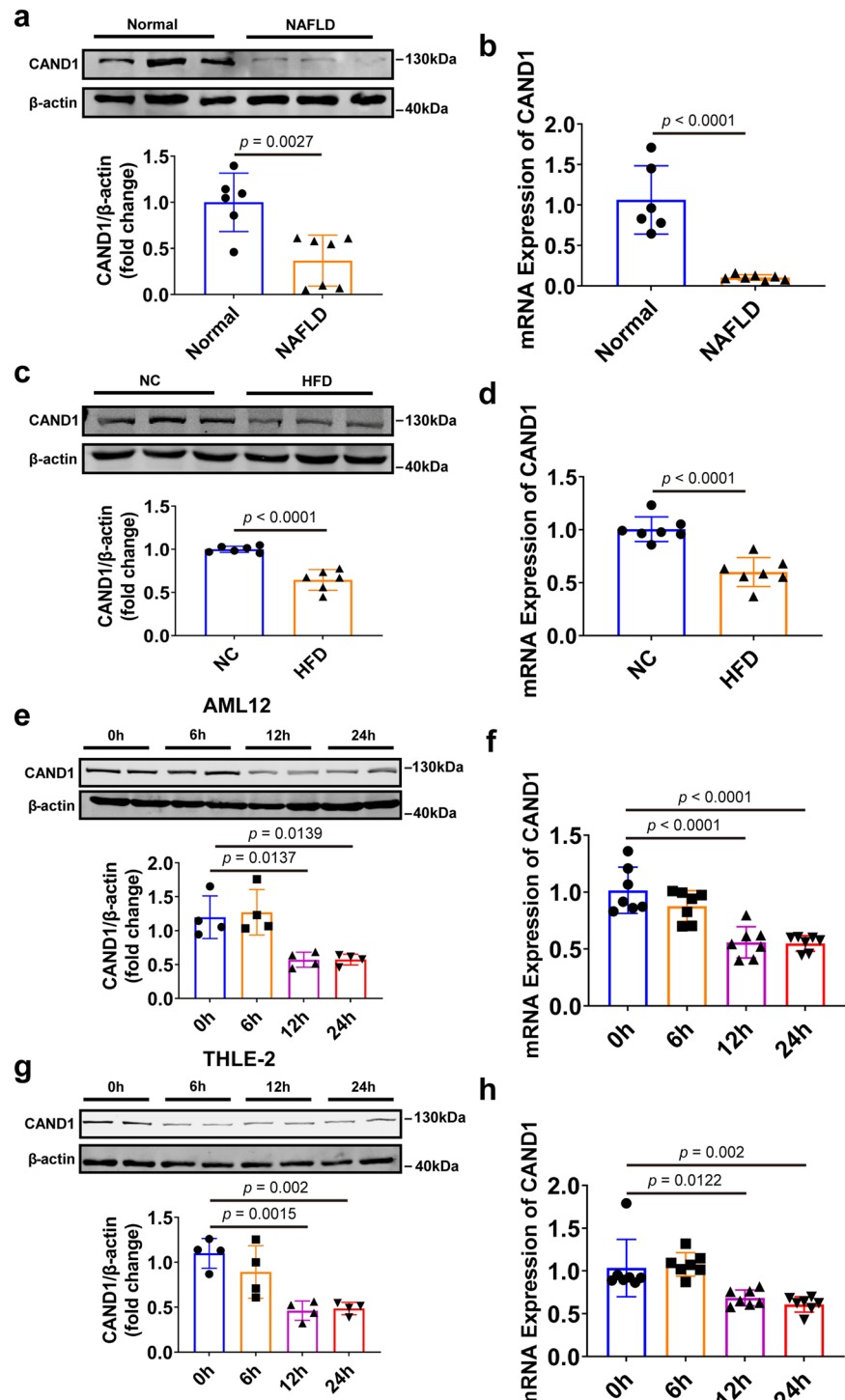

**Fig. 1 | CAND1 expression is decreased in the liver during hepatic steatosis.**
**a** Western blotting of CAND1 in liver samples from normal donors ($n = 6$) and NAFLD patients ($n = 7$). **b** CAND1 mRNA level in liver samples from normal donors ($n = 6$) and NAFLD patients ($n = 7$). **c** Western blotting of CAND1 in the livers from HFD-fed and NC-fed mice ($n = 6$). NC, Normal chow. **d** The mRNA levels of CAND1 in the livers from HFD-fed and NC-fed mice ($n = 7$). **e** Western blotting of CAND1 in

AML12 cells treated with PA ($n = 4$). **f** CAND1 mRNA level in AML12 cells treated with PA ($n = 7$). **g** Western blotting of CAND1 in THLE-2 cells treated with PA ($n = 4$). **h** CAND1 mRNA level in THLE-2 cells treated with PA ($n = 7$). $p$ values obtained via two-tailed unpaired Student's $t$ tests, one-way ANOVA with Tukey's multiple comparisons test. The data were shown as means ± SD of independent biological replicates. Source data are provided as a Source data file.

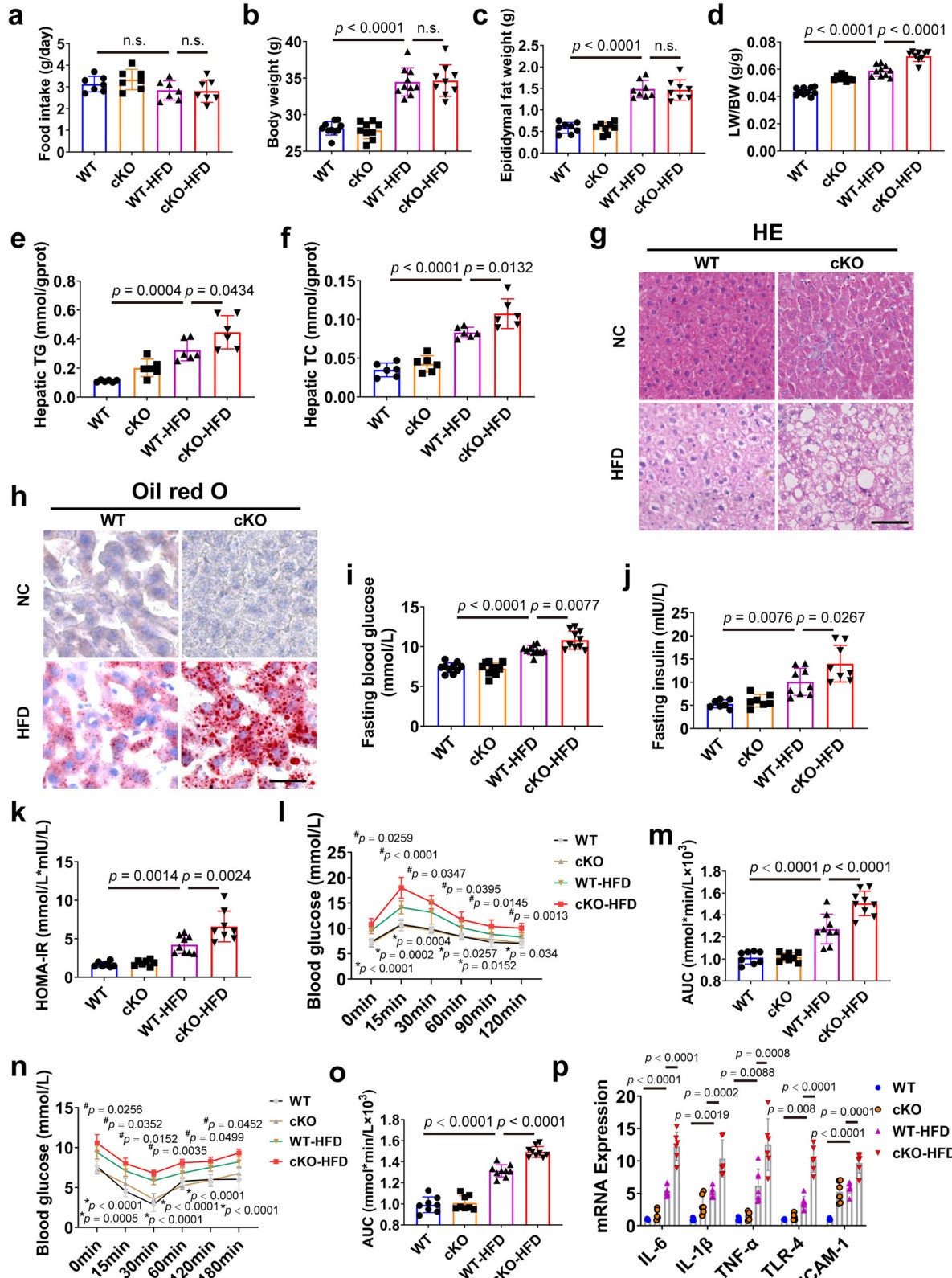

We further investigated the effect of CAND1 on NAFLD in female mice. WT-HFD and CAND1 cKO-HFD female mice had comparable food intake and body weight (Supplementary Fig. 7a, b). The LB/BW ratio was also increased in CNAD1 cKO-HFD female mice (Supplementary Fig. 7c). Moreover, lipid accumulation was more severe in the livers of CAND1 cKO-HFD than WT-HFD female mice (Supplementary Fig. 7d–f). These results indicate that CAND1

deficiency in hepatocytes also aggravates HFD-induced hepatic steatosis in female mice.

## Hepatocyte-specific CAND1 overexpression ameliorates liver injury after HFD treatment

We further established hepatocyte-specific CAND1 overexpression (conditional knockin, cKI) male mouse lines to validate the function of

**Fig. 2 | Hepatocyte-specific CAND1 deficiency exacerbates HFD-induced liver injury in male mice. a** Food intake of CAND1 cKO and WT male mice after 16 weeks of NC or HFD feeding ($n = 7$). **b** Body weight in indicated groups (WT, $n = 10$; cKO, $n = 9$; WT-HFD, $n = 10$; cKO-HFD, $n = 9$). **c** Epididymal fat weight ($n = 8$). **d** LW/BW ratio (WT, $n = 10$; cKO, $n = 9$; WT-HFD, $n = 10$; cKO-HFD, $n = 9$). **e, f** TG and TC content in indicated groups ($n = 6$). **g** H&E staining of liver sections ($n = 8$). Scale bar=50 μm. Magnification 200×. **h** Oil red O staining of liver sections ($n = 8$). Scale bar = 50 μm. Magnification 200×. **i** Fasting blood glucose ($n = 10$). **j** Fasting insulin levels (WT, $n = 7$; cKO, $n = 7$; WT-HFD, $n = 9$; cKO-HFD, $n = 8$). **k** HOMA-IR index (WT, $n = 7$; cKO, $n = 7$; WT-HFD, $n = 9$; cKO-HFD, $n = 8$). **l** GTTs (WT, $n = 8$; cKO, $n = 8$; WT-HFD, $n = 9$; cKO-HFD, $n = 9$). **m** Areas under the curve (AUC) of GTT (WT, $n = 8$; cKO, $n = 8$; WT-HFD, $n = 9$; cKO-HFD, $n = 9$). **n** ITTs (WT, $n = 8$; cKO, $n = 8$; WT-HFD, $n = 9$; cKO-HFD, $n = 9$). **o** AUC of ITT (WT, $n = 8$; cKO, $n = 8$; WT-HFD, $n = 9$; cKO-HFD, $n = 9$). **p** mRNA levels of inflammatory cytokines in the livers (WT, $n = 5$; cKO, $n = 6$; WT-HFD, $n = 7$; cKO-HFD, $n = 6$). $p$ values obtained via one-way ANOVA with Tukey's multiple comparisons test. The data were shown as means ± SD of independent biological replicates. Source data are provided as a Source data file. LW/BW, liver weight/body weight. TG, Triglyceride. TC, Total cholesterol. n.s., not significant.

hepatocyte CAND1 on liver injury after HFD treatment (Supplementary Fig. 5a, b). There was no statistical difference in food intake between WT and CAND1 cKI mice after HFD or NC treatment (Fig. 3a). HFD treatment increased body weight and epididymal fat weight of WT and CAND1 cKI male mice after 16 weeks of HFD, and the body weight and epididymal fat weight of WT-HFD and cKI-HFD mice were comparable (Fig. 3b, c). In contrast to the alterations observed in CAND1 cKO mice, hepatocyte-specific overexpression of CAND1 substantially protected against HFD-induced hepatic steatosis, hepatic resistance and inflammation. Specifically, overexpression of CAND1 in hepatocytes ameliorated hepatic steatosis, and decreased LW/BW ratio and liver contents of TG and TC (Fig. 3d–f). HFD treatment-induced lipid accumulation was decreased in the livers of CAND1 cKI (Fig. 3g, h). Glucose tolerance and insulin sensitivity was greatly improved in CAND1 cKI-HFD than WT-HFD mice (Fig. 3i–o). Moreover, mRNA levels of pro-inflammatory factors were lower in CAND1 cKI-HFD than WT-HFD mice (Fig. 3p). Although, Sirius red staining revealed that there was no difference in fibrotic area between WT-HFD and cKI-HFD male mice (Supplementary Fig. 6b). These data confirmed the protective role of CAND1 in HFD-induced liver injury.

### CAND1 decreases the formation of Cullin1, FBXO42 and ACAA2 complexes, and prevents ubiquitination and degradation of ACAA2

CAND1 controls the dynamics of global CRLs network by regulating the assembly of F-box containing substrate receptors and Cullin1 complexes[21]. To elucidate the mechanism by which CAND1 regulates the development of NAFLD, we performed Co-IP and Mass Spectrometry analysis to identify Cullin1-binding proteins in PA treated L02 cells transfected with NC or siCAND1. A total of 72 deregulated Cullin1-binding proteins were obtained, with 39 up-regulated genes and 33 down-regulated (Fig. 4a, b). Among the up-regulated genes, acetyl-CoA acyltransferase 2 (ACAA2) and F-box protein 42 (FBXO42) drew our attention (Fig. 4b). ACAA2 is a thiolytic enzyme for fatty acid β-oxidation, overexpression of which promotes the oxidation of fatty acids and inhibits lipid accumulation[22,23]. The F-box protein FBXO42 is a substrate receptor, which were shown to recruit target proteins and promote their ubiquitinated degradation[24,25]. We speculated that Cullin1, FBXO42, and ACAA2 may form a SCF complex to regulate the ubiquitination and degradation of ACAA2 and thus the effects of CAND1 on NAFLD. Molecular docking also predicted that FBXO42 binds to ACAA2 and Cullin1 (Fig. 4c).

We examined Cullin1, FBXO42, and ACAA2 complexes formation in AML12 and THLE-2 cells. The formation of ACAA2, Cullin1 and FBXO42 complex was markedly increased in siCAND1 compared to NC group after PA treatment (Fig. 4d, e), and ACAA2 protein expression was significantly downregulated in siCAND1 group compared with NC group after PA treatment (Fig. 4d, e).

ACAA2 protein expression was noticeably reduced in WT-HFD mice compared to WT mice (Fig. 4f, h). Compared with WT-HFD mice, ACAA2 protein was significantly decreased in CAND1 cKO-HFD mice, indicating higher ubiquitination of ACAA2 (Fig. 4f, g). In CAND1 cKI-HFD male mice, ACAA2 protein was significantly upregulated and its ubiquitination level was obviously decreased (Fig. 4h, i).

We next assayed Cullin1, FBXO42, and ACAA2 complexes by Co-IP assay. The formation of ACAA2, Cullin1, and FBXO42 complex was significantly increased in WT-HFD mice compared to WT mice, and it was more abundant in CAND1 cKO-HFD than WT-HFD mice (Fig. 5a). However, the ACAA2, Cullin1, and FBXO42 complex was less abundant in CAND1 cKI-HFD than WT-HFD mice (Fig. 5b). These results suggest that CAND1 suppresses the assembly of ACAA2, Cullin1 and FBXO42 complexes, thereby inhibiting the ubiquitinated degradation of ACAA2.

### CAND1 regulates the development of NAFLD through ACAA2

To further clarify whether the regulatory effects of CAND1 on NAFLD was mediated by ACAA2, we injected AAV8-ACAA2 virus to male mice by tail vein to specifically increase the expression of ACAA2 in hepatocytes (Supplementary Fig. 8a, b). Overexpression of ACAA2 greatly improved the pathological alterations of CAND1 cKO-HFD mice, including LB/BW ratio, hepatic TG and TC levels, and lipid accumulation (Fig. 6a–e). In addition, ACAA2 overexpression significantly abolished the aggravating effects of CAND1 conditional knockout on fasting glucose, fasting insulin, and HOMA-IR index of mice fed with HFD (Fig. 6f–h). Furthermore, ACAA2 overexpression improved glucose tolerance and insulin sensitivity of CAND1 cKO-HFD mice, evidenced by intraperitoneal GTT and ITT (Fig. 6I–L). ACAA2 overexpression also reduced the levels of pro-inflammatory factors IL-6, TNF-α, and TLR-4 in CAND1 cKO-HFD mice (Fig. 6M). These results indicated that the effects of hepatocyte CAND1 on NAFLD progression were profoundly mediated by ACAA2.

### Hepatocyte-specific CAND1 deficiency promotes starvation-induced fatty liver

Under fasting conditions, excessive fatty acids are re-esterified into triglyceride for intracellular storage in the liver, which leads to starvation-induced fatty liver[26,27]. We then explored whether CAND1 also has a regulatory effect on starvation-induced fatty liver. After starving male mice for 24 h, we examined the accumulation of hepatic lipids. The results showed that liver lipid accumulation was more severe after starvation in CAND1 cKO than WT male mice (Fig. 7a, b). The levels of TG and TC in the liver were also significantly increased in CAND1 cKO male mice (Fig. 7c, d). Moreover, the level of ACAA2 was significantly decreased in CAND1 cKO compared to WT mice (Fig. 7e). These data indicated that CAND1 also inhibits starvation-induced liver lipid metabolism.

### Androgen receptor regulates CAND1 transcription

Since the mRNA level of CAND1 was reduced in the liver of NAFLD, we then explored the transcriptional regulation of CAND1. We predicted the transcription factors of *CAND1* gene using the JASPAR website (https://jaspar.genereg.net/) and identified four potential binding sites of androgen receptor (AR) on the promoter region of *CAND1* gene (Fig. 8a, Supplementary Table 1). We observed a decrease in testosterone and AR levels in NAFLD male mice compared to normal chow male mice (Fig. 8b, c). AR expression was also decreased in AML12 cells after PA induction (Fig. 8d). Interestingly, AR expression and testosterone levels were significantly reduced in CAND1 cKO male mice compared to WT male mice (Supplementary Fig. 9a, b).

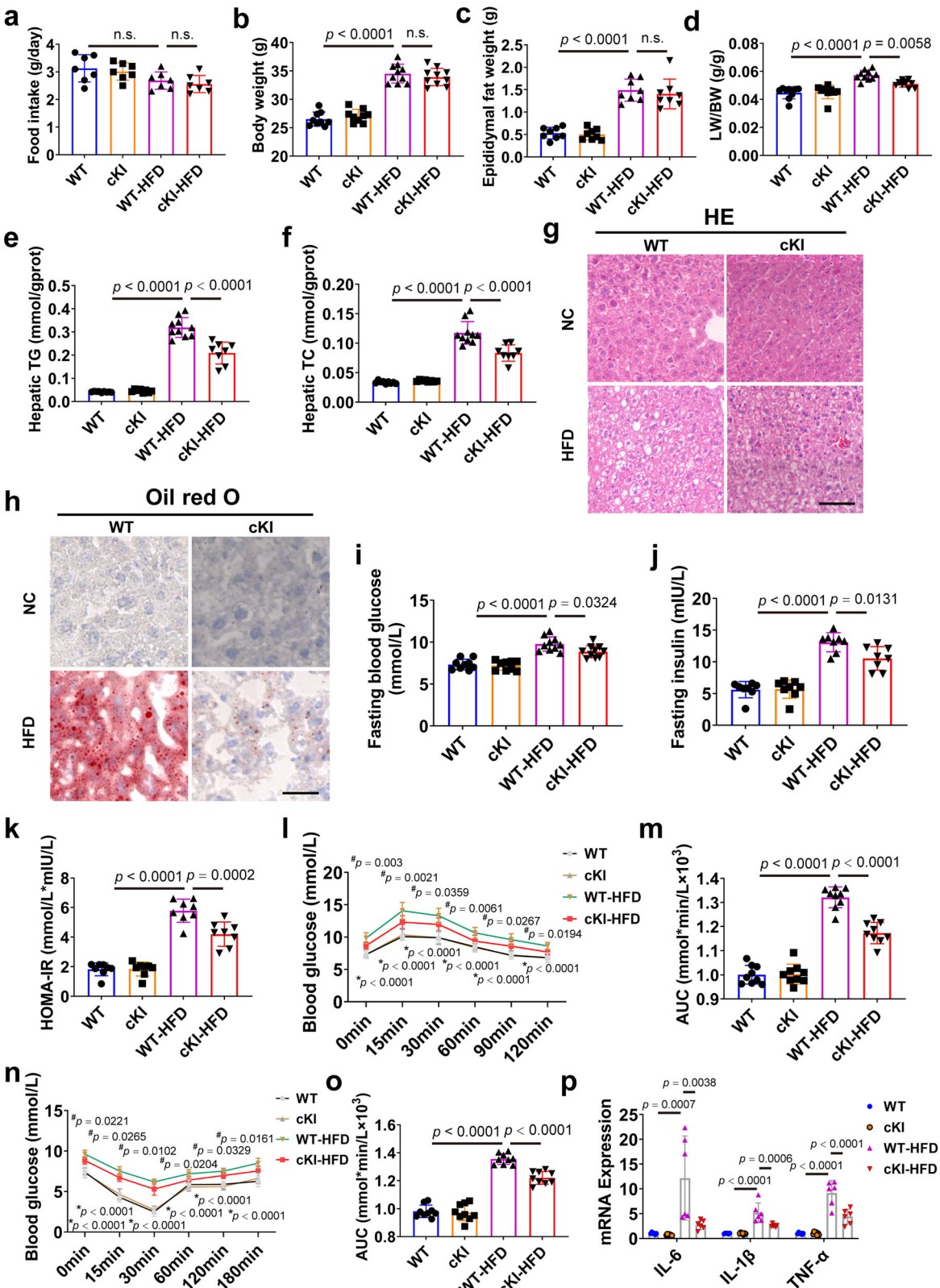

To clarify the direct effects of AR on *CAND1* transcription, we co-transfected AR overexpressing plasmid and plasmids containing a sequentially truncated *CAND1* promoter into cells. Four putative AR binding sites were identified in the *CAND1* promoter region. Sequential deletions of these binding sites revealed that AR-binding site 1 (-187 upstream of the transcription start site) is the major one for AR-facilitated *CAND1* transcriptional activity (Fig. 8e). In addition,

testosterone treatment upregulated the mRNA and protein levels of CAND1 in AML12 cells (Fig. 8f, g). PA-induced lipid accumulation was similar in siCAND1 group and siAR group (Fig. 8h–j). Concomitant knockdown of AR and CAND1 exacerbated PA-induced lipid accumulation compared with CAND1 knockdown alone in AML12 cells (Fig. 8h–j, Supplementary Fig. 9c), while overexpression of CAND1 attenuated AR deficiency-caused lipid accumulation in AML12 cells

**Fig. 3 | Hepatocyte-specific CAND1 overexpression attenuates HFD-induced liver injury in male mice. a** Food intake of hepatocyte-specific conditional knockin (cKI) and WT male mice after 16 weeks of NC or HFD feeding (*n* = 7). **b** Body weight in indicated groups (WT, *n* = 9; cKI, *n* = 9; WT-HFD, *n* = 10; cKI-HFD, *n* = 10). **c** Epididymal fat weight (*n* = 8). **d** LW/BW ratio (WT, *n* = 9; cKI, *n* = 9; WT-HFD, *n* = 10; cKI-HFD, *n* = 10). **e**, **f** TG and TC content in the indicated groups (WT, *n* = 9; cKI, *n* = 8; WT-HFD, *n* = 10; cKI-HFD, *n* = 8). **g** H&E staining of liver sections (*n* = 8). Scale bar = 50 μm. Magnification 200×. **h** Oil red O staining of liver sections (*n* = 8). Scale

bar = 50 μm. Magnification 200×. **i** Fasting blood glucose (WT, *n* = 9; cKI, *n* = 9; WT-HFD, *n* = 10; cKI-HFD, *n* = 10). **j** Fasting insulin levels (*n* = 8). **k** HOMA-IR index (*n* = 8). **l** GTTs (*n* = 9). **m** AUC of GTT (*n* = 9). **n** ITTs (*n* = 9). **o** AUC of ITT (*n* = 9). **p** mRNA levels of inflammatory cytokines in the livers (WT, *n* = 6; cKI, *n* = 7; WT-HFD, *n* = 6; cKI-HFD, *n* = 6). *p* values obtained via one-way ANOVA with Tukey's multiple comparisons test. The data were shown as means ± SD of independent biological replicates. Source data are provided as a Source data file. LW/BW, liver weight/body weight. TG, Triglyceride. TC, Total cholesterol. n.s., not significant.

after PA treatment (Fig. 8h–j). We administered AR antagonist enzalutamide (10 mg/kg) to male mice and found that the mRNA and protein levels of CAND1 were significantly reduced (Fig. 8k, l). Moreover, the level of CAND1 in the livers of mice increased from 2-month to 6-month after birth, and decreased at 12-month old, which was consistent with the expression change of AR (Fig. 8m–o). These results suggested that CAND1 transcription was modulated by androgen receptor.

## Discussion

In the current study, we discovered that CAND1 blocks the assembly of Cullin1, FBXO42, and ACAA2 complexes and inhibits ubiquitinated degradation of ACAA2, thereby maintaining the rate of fatty acid degradation in hepatocytes and reducing lipid accumulation in the liver. The androgen receptor determines the transcription of *CAND1* by binding to the -187 to -2000 promoter region. Our findings indicate that CAND1 is a promising target for the treatment of NAFLD.

As a precise way of tuning protein modification, ubiquitin molecules can accurately and selectively determine protein fate. The ubiquitination process precisely pinpoints the destination of proteins, which is essential for maintaining the stability of the intracellular environment. Disturbance of protein ubiquitination and degradation occurs in hepatocytes during the development of NAFLD[28–30]. CAND1 functions as an exchange factor to regulate the Skp1-Cullin1-F-box protein (SCF) repertoire to control the ubiquitination and degradation of multiple proteins in cells[31]. Studies showed that CAND1 deficiency disrupted original degradation trajectory of proteins, resulting in an imbalance in homeostasis[14]. CAND1 knockout contributed to rapid proliferation and high migration of lung cancer cells[17]. CAND1 deficiency impaired the assembly of Cullin1/atrogin1/calcineurin complex and inhibited the ubiquitination and degradation of calcineurin, and aggravated cardiac hypertrophic phenotypes in cardiac hypertrophy[18]. In this study, we found CAND1 is also a critical regulator of NAFLD. It was significantly reduced in the liver of NAFLD male patients and male mice. Hepatocyte-specific CAND1 knockout male mice showed increased steatosis in HFD-treatment and fasting state, while CAND1 overexpression male mice manifested the reverse phenotype. Insulin resistance is a profound characteristic of NAFLD and can form a vicious cycle with inflammation and fatty acid accumulation in the liver in both humans and rodents[32]. Inflammation is a major contributor in the etiology of insulin resistance and hepatic steatosis[33]. Consistently, we found CAND1 significantly ameliorated insulin sensitivity and glucose tolerance and inhibited pro-inflammatory cytokine production in male mice. NAFLD is a sexual dimorphic disease and the prevalence and severity of NAFLD are affected by sex[34]. In adult populations, NAFLD prevalence is higher in men than in pre-menopausal women[35]. Therefore, only NAFLD male patients were selected for analysis in GEO data. However, the regulation of CAND1 on NAFLD is not sexual dimorphic, as deletion of CAND1 also exacerbated NAFLD development in female mice.

Cullin-RING ubiquitin ligases (CRLs) determine the substrate specificity of ubiquitination reactions, and substrates are recruited to the Cullin core by binding to their cognate substrate receptor modules[36]. The presence of CAND1 ensures that the repertoire of CRLs remains dynamic, and when specific substrates emerge, it can

stabilize their cognate CRLs to maintain effective degradation, achieve a dynamic balance, and maintain the vitality of the microenvironment[21]. CAND1 knockdown enhanced proteasomal degradation of three-way junction protein lunapark (Lnp) and reduced the tubular ER network in mammalian cells[37]. Suppression of CAND1 expression specifically promoted the incorporation of F-box protein Skp2 into CRL1 complexes accompanied with a degradation of substrate protein p27[15,38]. The substrate receptor FBXO42 recruits target protein to the Cullin1-FBXO42 ubiquitin ligase complex. Higher level of FBXO42 and Cullin1 was observed in CAND1 and CAND2 double knockout cells[39]. In this study, we identified ACAA2 as a new substrate for FBXO42. Deletion of CAND1 increased the complexes of Cullin1/FBXO42/ACAA2 and promoted the degradation of substrate protein ACAA2.

ACAA2 is a thiolytic enzyme for the final step of fatty acid β-oxidation. In mammary epithelial cells, overexpression of ACAA2 inhibited triglyceride production[22]. Reduced expression of ACAA2 impaired fatty acid β-oxidation and eventually exacerbated kidney fibrosis in acute kidney injury[11]. Knockout ACAA2 homolog MTP reduced fatty acid oxidation capacity in the liver, and increased hepatic steatosis and the expression of inflammatory marker CD68, accelerating the progression of NAFLD[12]. Consistently, our study proved ACAA2 is a protective molecule in NAFLD. Hepatocyte-specific ACAA2 overexpression attenuated HFD-induced liver injury. Moreover, overexpression of ACAA2 rescued the detrimental effects of CAND1 deletion on NAFLD, implying the critical role of ACAA2 in mediating the function of CAND1.

However, considering the fact that CAND1 can regulate the expression of a large number of substrate proteins, we cannot rule out the involvement of other target substrates in the regulation of CAND1 on NAFLD. Solute Carrier Family 25 Member 24 (SLC25A24) is one of the substrates we found in the study. Deletion of SLC25A24 in female mice led to lower body weights, white adipose tissue weights and liver weight when consuming the high-fat diet[40]. It is possible that SLC25A24 may regulate the effects of CAND1 on NAFLD. Furthermore, increased neddylation of CUL3, an essential regulatory element of the NRF2 complex, by AGEs rendered NRF2 less stable and destined for degradation, which led to advanced glycation end-products clearance receptor (AGER1) downregulation and promoted NASH[41]. This cascade may be disturbed due to the loss of CAND1 in the early NAFLD. Therefore, the lack of deep study on multiple downstream targets of CAND1 is a limitation of the current study.

Androgen/androgen receptor (AR) signaling pathway plays an important role in stabilizing energy metabolism and resisting steatosis of hepatocytes and the development of NAFLD[42–45]. In this study, testosterone and androgen receptor levels were decreased in NAFLD models, which is correlated with the reduced expression of CAND1. Subsequently, we experimentally demonstrated that AR bound to the region from -187 to -2000 of *CAND1* promoter and enhanced *CAND1* transcription. Intervention in Androgen/AR pathway led to decreased CAND1 expression. Concomitant knockdown of AR and CAND1 exacerbated lipid accumulation compared to knockdown of CAND1 alone in AR-positive AML12 cells, while overexpression of CAND1 attenuated AR deficiency-caused lipid accumulation in AR-positive AML12 cells. These results firmly

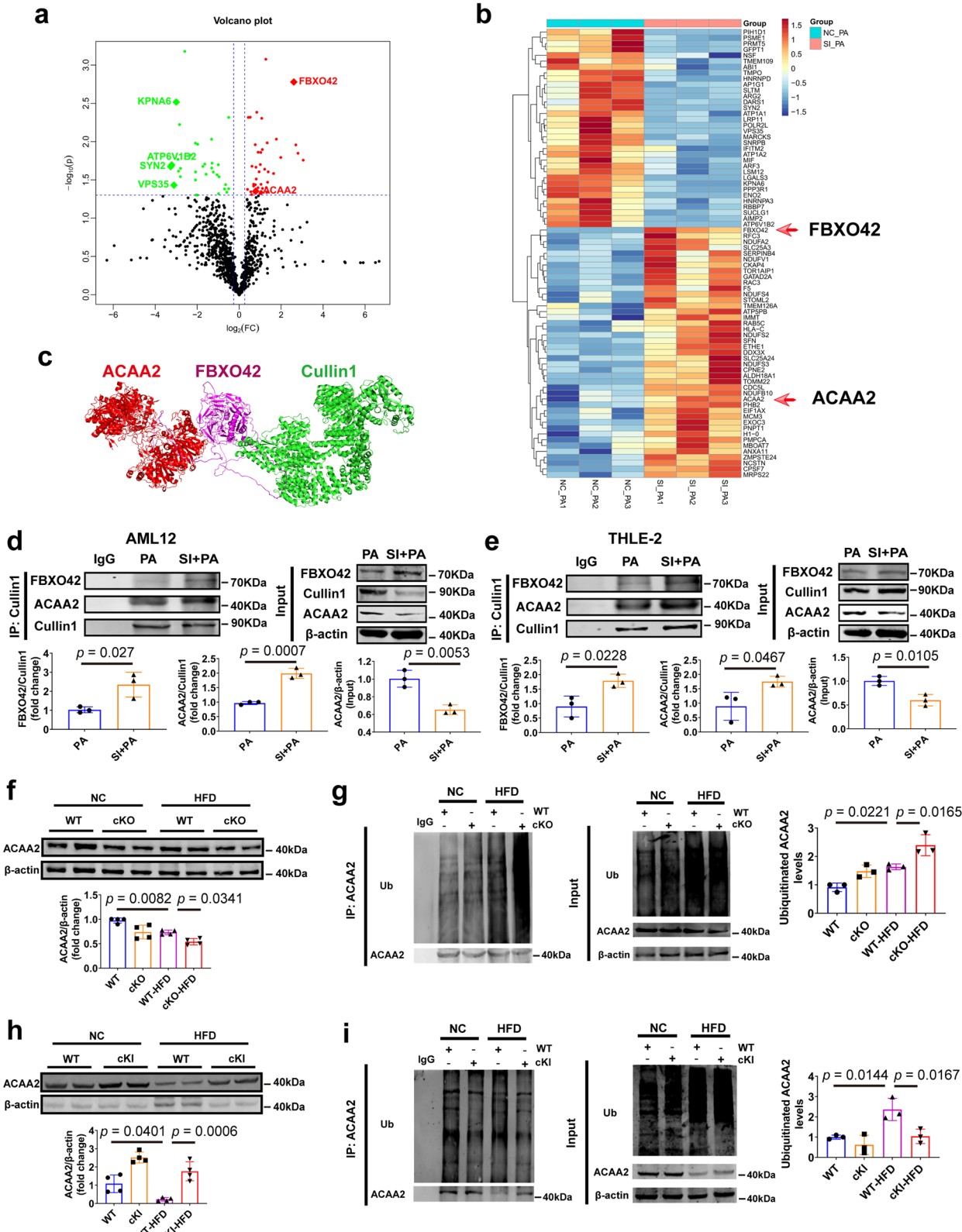

**Fig. 4 | CAND1 controls the ubiquitination and degradation of fatty acid β-oxidation gene ACAA2. a** Volcano plot of Cullin1-binding protein in L02 induced by PA with NC or siCAND1. **b** Heatmap of deregulated binding proteins in L02 cells induced by PA after knockdown of CAND1. **c** Molecular docking of Cullin1, FBXO42 and ACAA2. **d** Immunoprecipitated with anti-Cullin1 antibody of NC or SI CAND1 groups after PA treatment in AML12, and then immunoblotted with antibodies specific for FBXO42 and ACAA2 (n = 3). **e** Immunoprecipitated with anti-Cullin1 antibody of NC or SI CAND1 groups after PA treatment in THLE-2, and then immunoblotted with antibodies specific for FBXO42 and ACAA2 (n = 3). **f** Western blotting

of ACAA2 in the livers of WT and cKO mice after 16 weeks of NC or HFD feeding (n = 4). **g** Ubiquitination of ACAA2 in the livers of WT and cKO mice after 16 weeks of NC or HFD feeding (n = 3). **h** Western blotting of CAND1 in the livers of WT and cKI mice after 16 weeks of NC or HFD feeding (n = 4). **i** Ubiquitination of ACAA2 in the livers of WT and cKI mice after 16 weeks of NC or HFD feeding (n = 3). p values obtained via two-tailed unpaired Student's t tests, one-way ANOVA with Tukey's multiple comparisons test. The data were shown as means ± SD of independent biological replicates. Source data are provided as a Source data file.

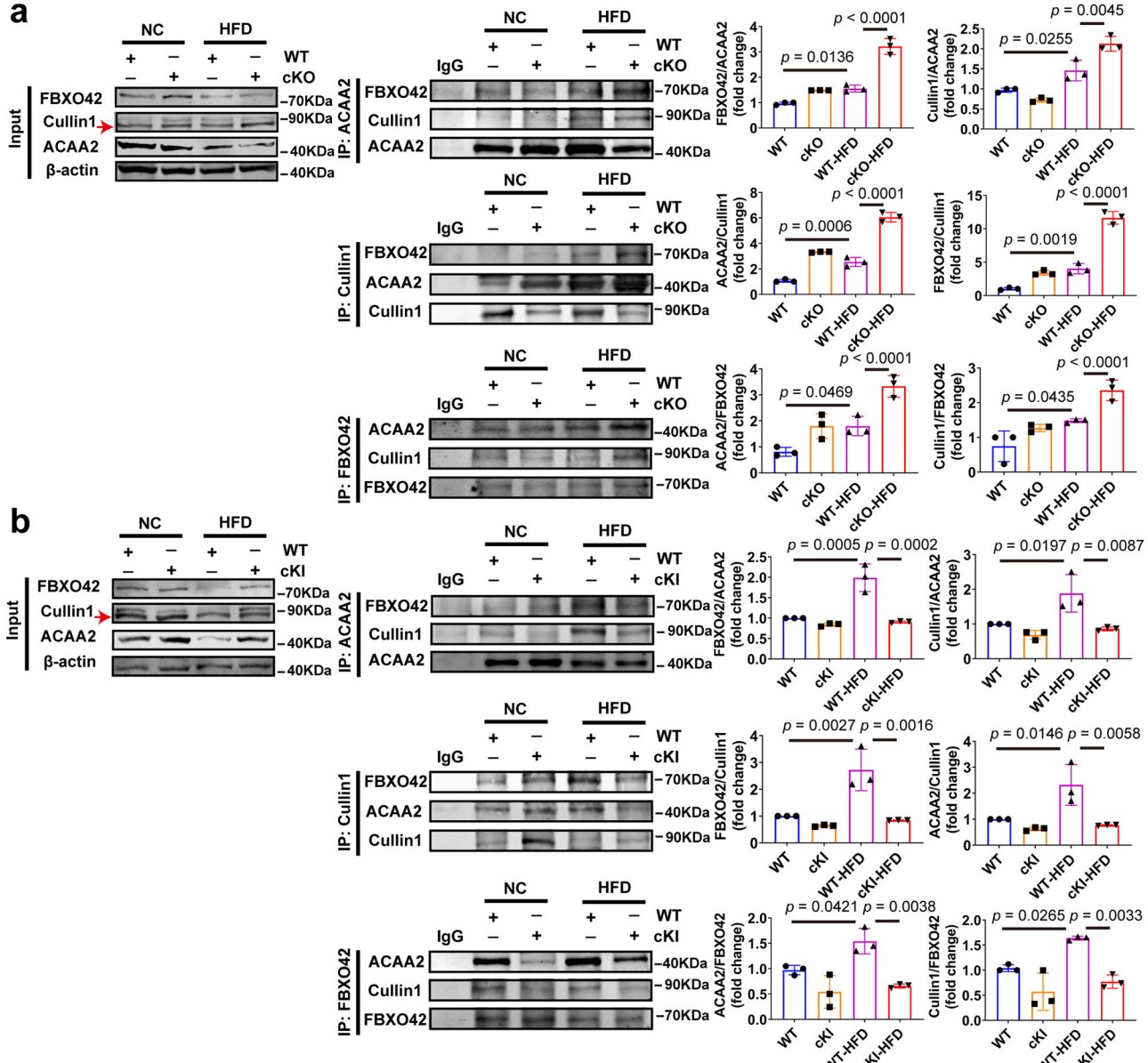

**Fig. 5 | CAND1 restrains the formation of Cullin1, FBXO42 and ACAA2 complexes. a** Livers of WT and cKO mice after 16 weeks of NC or HFD-feeding immunoprecipitated with anti-ACAA2/Cullin1/FBXO42 antibodies and then immunoblotted with antibodies specific for ACAA2/FBXO42/Cullin1, and quantification of the relative ACAA2/FBXO42/Cullin1 level (*n* = 3). **b** Livers of WT and cKI mice after 16 weeks of NC or HFD-feeding immunoprecipitated with anti-ACAA2/ Cullin1/FBXO42 antibodies and then immunoblotted with antibodies specific for ACAA2/FBXO42/Cullin1, and quantification of the relative ACAA2/FBXO42/Cullin1 level (*n* = 3). *p* values obtained via one-way ANOVA with Tukey's multiple comparisons test. The data were shown as means ± SD of independent biological replicates. Source data are provided as a Source data file.

support the important role of AR-CAND1 axis in NAFLD development. Testosterone level is high in adolescence and markedly decrease in old age[46,47]. We found that androgen receptor was increased at the age of 6-month and declined at the age of 12-month in mice. The same pattern of expression change of CAND1 was demonstrated. This observation further supports the direct regulation of androgen receptor on CAND1 expression and implies the key role of CAND1 in increased susceptibility to NAFLD of older people. Interestingly, the levels of testosterone and hepatic AR expression were reduced in cKO male mice, which provided additional evidence to support reciprocal feedback regulation relationship between CAND1 and AR in male mice. Given the regulatory effect of CAND1 on lipid accumulation in female mice and AR-negative cell lines (THLE-2 and HepG2), there may be other

pathways regulating CAND1 expression and involved in the development of NAFLD in female mice.

In conclusion, CAND1 is a negative regulator of NAFLD, which exerts its action by restraining the assembly of Cullin1, FBXO42 and ACAA2 complexes and inhibiting ubiquitinated degradation of ACAA2 (Fig. 9). The findings suggest that enhancing the function of CAND1 is a promising strategy for the development of a therapeutic agent for NAFLD.

## Methods
### Human liver samples
All procedures that involved human samples were approved by Harbin Medical University (Harbin, China; project license number: IRB5025722). NAFLD liver samples were obtained from male

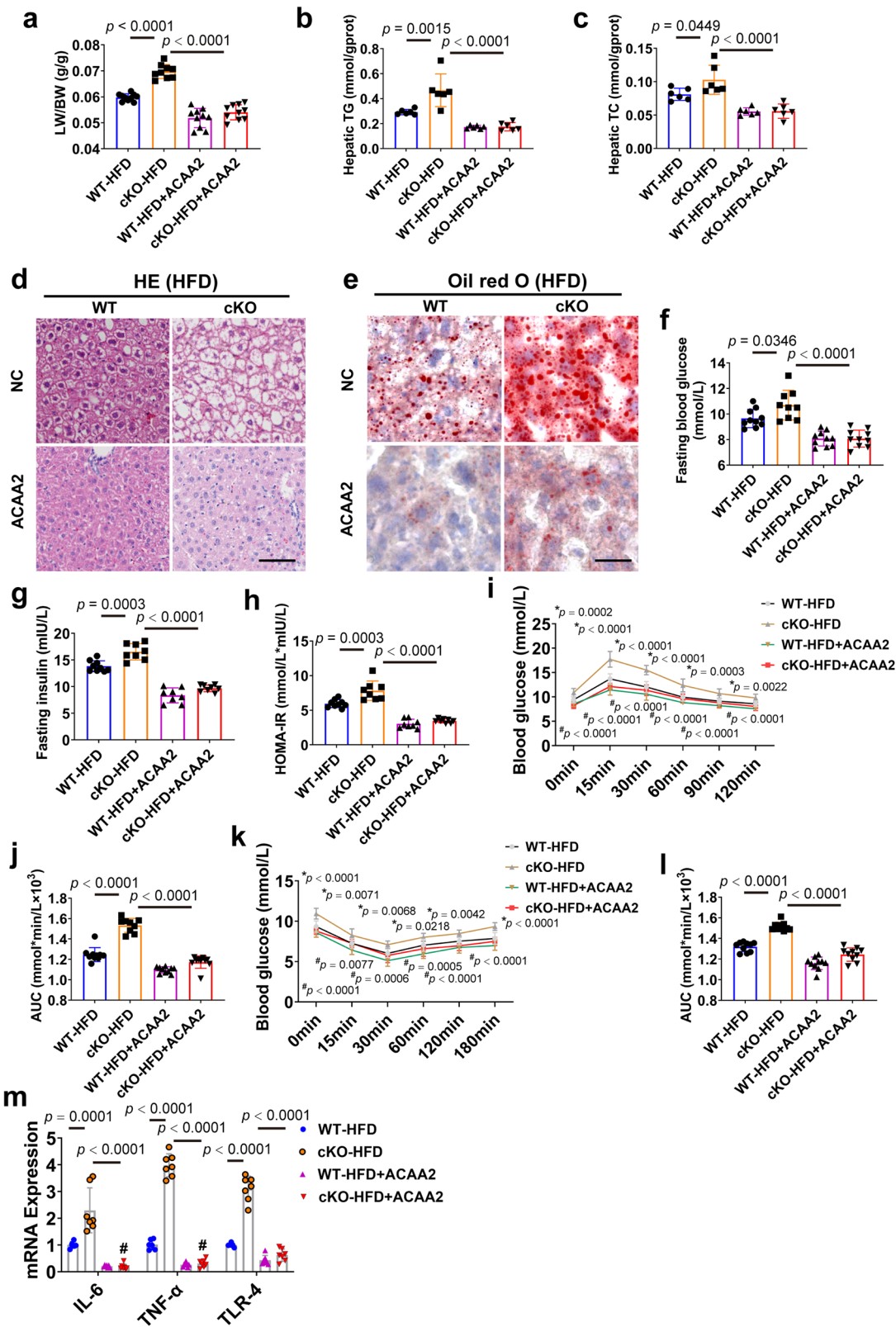

patients with hepatic steatosis who had undergone liver biopsy or transplantation. Control liver samples were collected from healthy areas of male donor livers undergoing hepatectomy for hepatic cysts. Informed consent was obtained from all participants. The general characteristics of the NAFLD human liver samples are listed in Supplementary Table 2.

**Animals and treatment**

All experiments complied with the guiding principles for the care and use of laboratory animals in Harbin Medical University and were approved by the Ethics Committee for Animal Experimentation of the School of Pharmacy, Harbin Medical University (project license number: IRB5025722). Male adult mice (8 weeks old) were used in the

**Fig. 6 | ACAA2 overexpression ameliorates CAND1 deficiency-potentiated hepatic steatosis, insulin resistance and inflammation in male mice. a** LW/BW ratio in the indicated groups after 16 weeks of HFD feeding (WT-HFD, $n = 10$; cKO-HFD, $n = 9$; WT-HFD + ACAA2, $n = 10$; cKO-HFD + ACAA2, $n = 10$). **b, c** TG and TC content in the indicated group after 16 weeks of HFD feeding ($n = 6$). **d** H&E staining of liver sections in the indicated groups ($n = 8$). Scale bar = 50 μm. Magnification 200×. **e** Oil red O of liver sections staining ($n = 8$). Scale bar=50 μm. Magnification 200×. **f** Fasting blood glucose (WT-HFD, $n = 10$; cKO-HFD, $n = 9$; WT-HFD + ACAA2, $n = 10$; cKO-HFD + ACAA2, $n = 10$). **g** Fasting insulin levels (WT-HFD, $n = 9$; cKO-HFD, $n = 8$; WT-HFD + ACAA2, $n = 8$; cKO-HFD + ACAA2, $n = 8$). **h** HOMA-IR index (WT-HFD, $n = 9$; cKO-HFD, $n = 8$; WT-HFD + ACAA2, $n = 8$; cKO-HFD + ACAA2, $n = 8$).

**i** GTTs (WT-HFD, $n = 10$; cKO-HFD, $n = 9$; WT-HFD + ACAA2, $n = 10$; cKO-HFD + ACAA2, $n = 10$). **j** AUC of GTT (WT-HFD, $n = 10$; cKO-HFD, $n = 9$; WT-HFD + ACAA2, $n = 10$; cKO-HFD + ACAA2, $n = 10$). **k** ITTs (WT-HFD, $n = 10$; cKO-HFD, $n = 9$; WT-HFD + ACAA2, $n = 10$; cKO-HFD + ACAA2, $n = 10$). **l** AUC of ITT (WT-HFD, $n = 10$; cKO-HFD, $n = 9$; WT-HFD + ACAA2, $n = 10$; cKO-HFD + ACAA2, $n = 10$). **m** mRNA levels of inflammatory cytokines in the livers of the indicate group after 16 weeks of HFD feeding (WT-HFD, $n = 6$; cKO-HFD, $n = 7$; WT-HFD + ACAA2, $n = 7$; cKO-HFD + ACAA2, $n = 7$). $p$ values obtained via one-way ANOVA with Tukey's multiple comparisons test. The data were shown as means ± SD of independent biological replicates. Source data are provided as a Source data file. LW/BW, liver weight/body weight. TG, Triglyceride. TC, Total cholesterol.

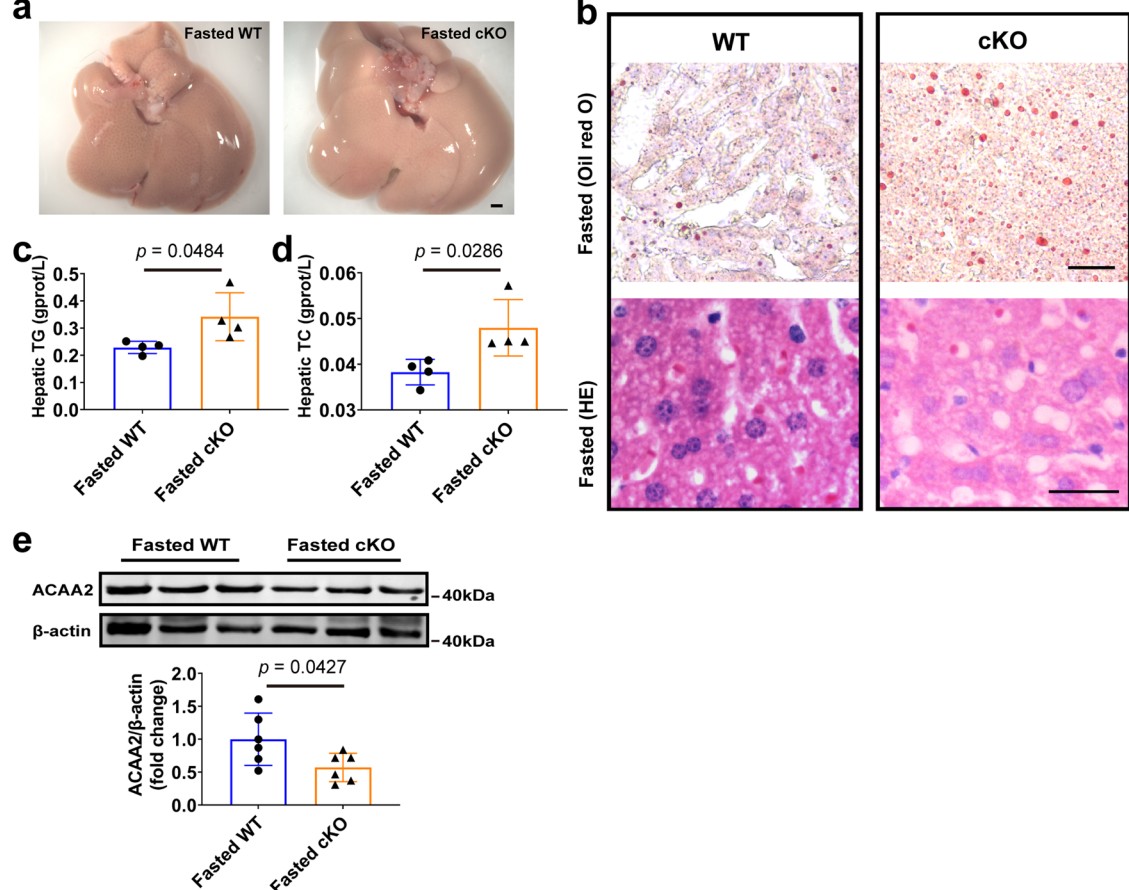

**Fig. 7 | Hepatocyte-specific CAND1 knockout promotes starvation-induced fatty liver in male mice. a** Gross images of liver and H&E staining from WT and cKO male mice that were fasted for 24 h ($n = 5$). Scale bar=2 mm. Magnification 2×. **b** Oli red O and H&E staining from WT and cKO male mice that were fasted for 24 h ($n = 5$). Scale bar=50 μm. Oil red O magnification 200×, H&E magnification 400×.

**c, d** TG and TC content from WT and cKO male mice that were fasted for 24 h ($n = 4$). **e** Western blotting of ACAA2 ($n = 6$). $p$ values obtained via unpaired two-sided Student's $t$-test. The data were shown as means ± SD of independent biological replicates. Source data are provided as a Source data file. TG, Triglyceride. TC, Total cholesterol.

present study. Mice were housed in a facility with 12-h light/12-h dark cycle at 23 ± 3 °C and 30-70% humidity. Male mice were incorporated into the experiments and were continuously fed either a high-fat diet (HFD, D09100310, BioPike, Beijing, China) or a normal chow (NC) diet (D09100304, BioPike, Beijing, China). HFD contains 40 % Fat (Palm Oil), 40 % Carbohydrate and 20% protein for a total of 4.49 kcal per 1 g of diet. Normal chow contains 10 % Fat, 70 % Carbohydrate and 20% protein for a total of 3.85 kcal per 1 g of diet. Provide 5 g of food per mouse per day and replace new food the next day. CAND1 heterozygous male mice and cKO female mice were fed high-fat diet for 12 weeks. cKO and cKI male mice fed high-fat diet for 16 weeks. Mice were sacrificed by isoflurane inhalation in a closed chamber where the

effective gas concentration (5%) was reached within seconds for further study.

### Cell lines
AML12 and HepG2 cells were purchased from Procell Life Science& Technology Co., Ltd. THLE-2 cell was purchased from iCell Bioscience Inc. L02 cell was purchased from the Type Culture Collection of the Chinese Academy of Sciences, Shanghai, China. HepG2 and L02 cells were cultured in a standard medium comprising DMEM (C3110-0500; VivaCell), 20% FBS (04-001-LA; BI), and 1% penicillin–streptomycin (03-031-1B; BI). THLE-2 cells were cultured using bronchial epithelial cell growth media (BEGM Bullet Kit, Cat#CC3170, iCell Bioscience Inc,

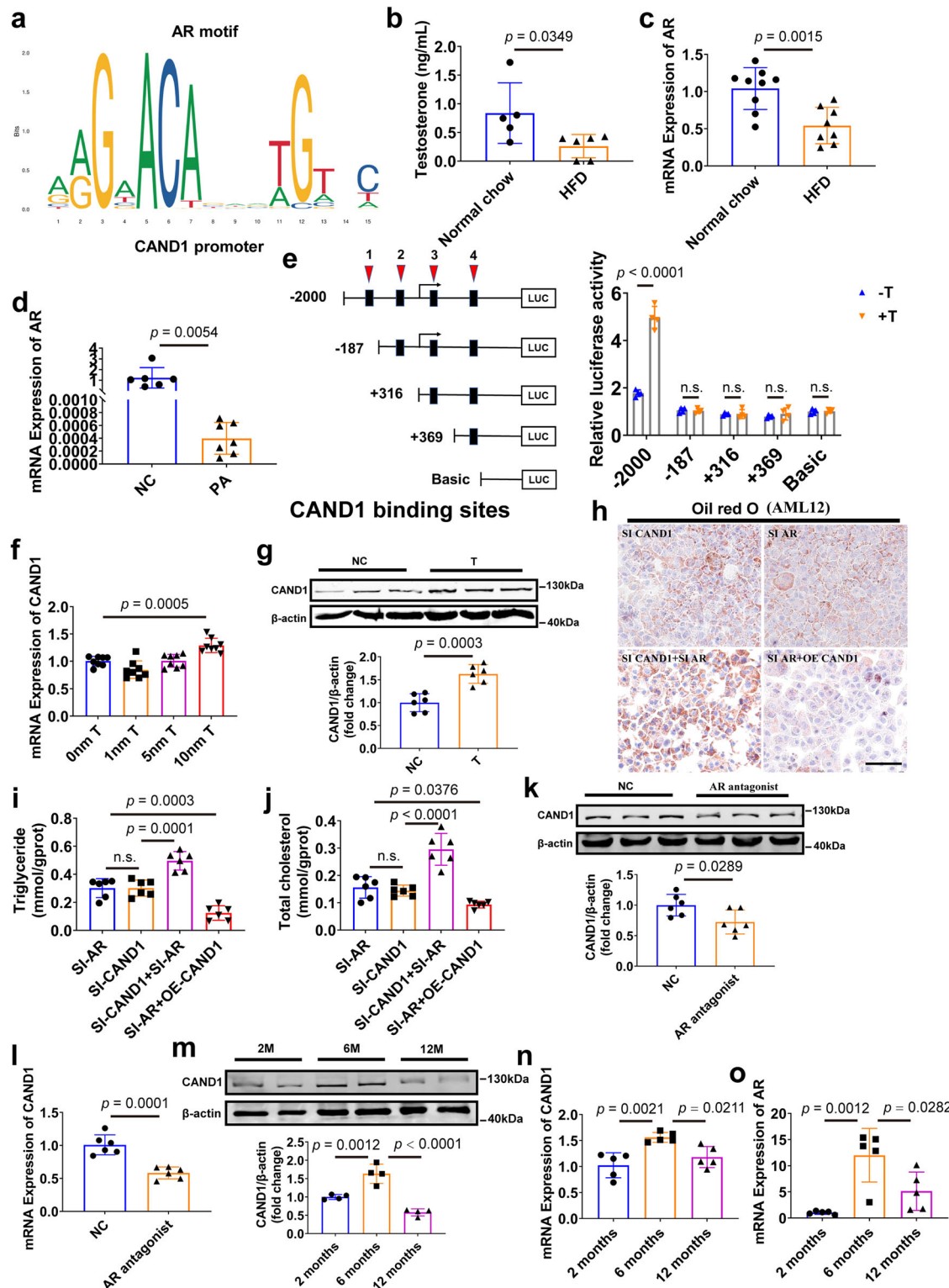

Shanghai, China). AML12 cells were cultured in a complete medium (CM-0602, Procell Life Science&Technology Co., Ltd, China). All cells were placed in 5% $CO_2$ under a water-saturated atmosphere in a cell incubator at 37 °C.

### Generation of genetically modified mice

Global CAND1 knockout mice were created using the CRISPR/Cas9 system by Cyagen Biosciences Inc (Guangzhou, China) as reported previously[18]. Two single guide RNAs (sgRNA1 and sgRNA2) targeting CAND1 exons 3 and 7 were designed (Supplementary Fig. 3a).

Cas9 mRNA and gRNA generated by in vitro transcription were then injected into the fertilized eggs for CAND1 knockout productions. The founders were genotyped by PCR followed by DNA sequencing analysis, and the positive founders were bred to the next generation which was further verified by PCR genotyping and DNA sequencing analysis. To genotype offspring, genomic PCR of tail DNA was performed with forward 5′-TGCCCTTCCCATCCTCATACCAG-3′ and reverse 5′- GGGAA ACACTTGCTGGAGTAGACTG-3′ primers.

CAND1[flox/flox] (CAND1-flox) mice were generated using the CRISPR/Cas9 system by Cyagen Biosciences Inc (Guangzhou, China). The third

**Fig. 8 | Androgen/Androgen receptor (AR) signaling regulates *CAND1* transcription. a** AR was predicted as a transcription factor of CAND1 by JASPAR. **b** Plasma testosterone levels of normal chow and HFD mice (Normal chow, $n = 5$; HFD, $n = 6$). **c** mRNA level of AR in the livers of normal chow and HFD mice (Normal chow, $n = 9$; HFD, $n = 8$). **d** mRNA level of AR in PA-treatment AML12 cells (NC, $n = 6$; PA, $n = 7$). **e** Serially truncated CAND1 promoter constructs and AR plasmids were cloned and transfected into cells. The relative luciferase activities were determined after testosterone treatment ($n = 4$). **f** mRNA level of CAND1 in AML12 cells treated with indicated concentrations of testosterone ($n = 8$). **g** Western blotting of CAND1 in AML12 cells treated with indicated concentrations (10 nM) of testosterone ($n = 6$). **h** Oil red O staining of SI-CAND1, SI-AR, SI-CAND1 + SI-AR and SI-AR + OE-

CAND1 groups after PA treatment ($n = 6$). Scale bar =50 μm. Magnification 200×. **i, j** TG and TC contents in the indicated group after PA treatment ($n = 6$). **k** CAND1 protein level in WT mice treated with AR antagonist, enzalutamide by western blot ($n = 6$). **l** CAND1 mRNA level in WT mice treated with AR antagonist, enzalutamide ($n = 6$). **m** CAND1 protein level in 2, 6 and 12 months old mice ($n = 4$). **n, o** CAND1 and AR mRNA levels in 2, 6, and 12 months old mice ($n = 5$). $p$ values obtained via two-tailed unpaired Student's $t$ tests, one-way ANOVA with Tukey's multiple comparisons test. The data were shown as means ± SD of independent biological replicates. Source data are provided as a Source data file. NC, negative control. n.s., not significant difference. TG, Triglyceride. TC, Total cholesterol.

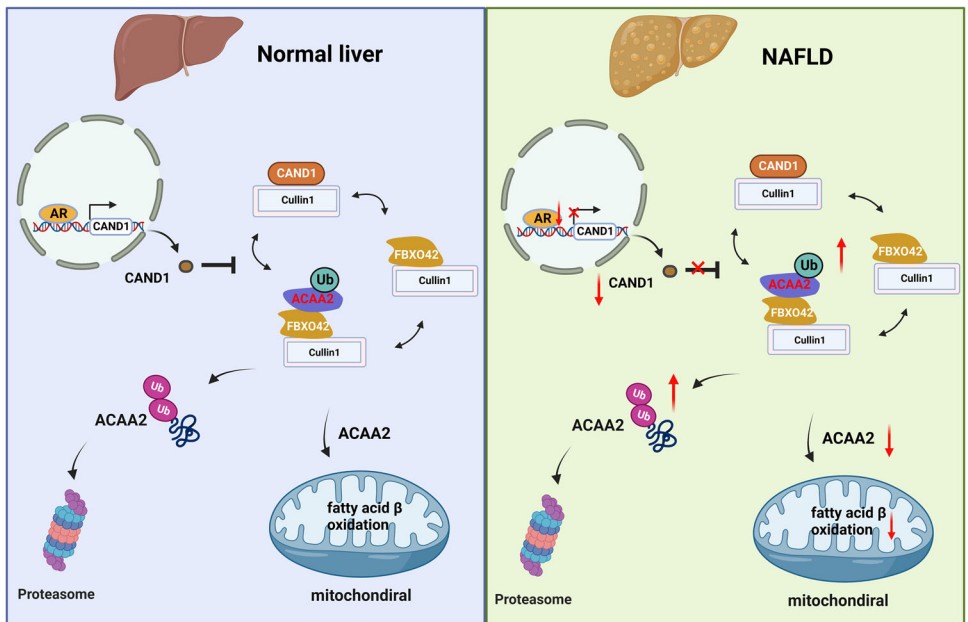

**Fig. 9 | Schematic diagram of the role of CAND1 in regulating ubiquitination and degradation of ACAA2 and NAFLD.** Under normal condition, AR regulates the transcription of CAND1 by binding to its promoter region. CAND1 finetunes the ubiquitination and degradation of ACAA2 via the Cullin1/Fbxo42/ACAA2 complex to maintain fatty acid metabolism homeostasis. In NAFLD, decreased AR expression

reduced CAND1 transcription. Reduction of CAND1 promotes the assembly of Cullin1, FBXO42, and ACAA2 complexes, and accelerates ubiquitinated degradation of ACAA2. which causes the insufficient β-oxidative capacity of fatty acids, accumulation of fatty acids, increased lipotoxicity in hepatocytes, and eventually NAFLD. The diagram was created using BioRender.

exon was flanked by loxP sites, and thus, two single guide RNAs (sgRNA1 and sgRNA2) targeting CAND1 exons 4 and 6 were designed. The donor vector containing exon 3 flanked by two loxP sites and the two homology arms were used as a template. A simple schematic diagram is shown in Supplementary Fig. 4a. The primers used for verifying flox gene in mice are: forward 5′-CGTGACTGACTGAAGCG-TAAACCTA-3′ and reverse 5′-GTCTACACTGAAAGCATCACAACTG-3′. CAND1$^{flox/flox}$ mice were then mated with Alb-cre mice, and homozygous mice with liver-specific knockout of CAND1 were obtained after multiple generations of breeding crosses.

CAND1 conditional knockin mice were generated by Cyagen Biosciences Inc (Guangzhou, China). The gRNA to mouse Cand1 gene, the donor vector containing "CAG promoter-loxP-PGK-Neo-6*SV40 pA-loxP-Kozak-Mouse Cand1 CDS-rBG pA" cassette, and Cas9 mRNA were co-injected into fertilized mouse eggs to generate targeted conditional knockin offspring (Supplementary Fig. 5a). F0 founder animals were identified by PCR followed by sequence analysis, which were bred to wildtype mice to test germline transmission and F1 animal generation. CAND1 conditional knockin mice were then mated with Alb-cre mice. Hepatocyte-specific CAND1 overexpression mice were obtained after multiple generations of reproductive crosses. All mice were compared only to non-transgenic or wild-type gender-matched littermates.

## Construction of ACAA2 overexpression virus

The recombinant serum type 8 adeno-associated virus system[48,49] (AAV8) was used to construct CAND1 gene overexpression virus (Genechem, Shanghai, China). Vector: GV704, element sequence: ApoE/hAATp-MCS-SV40 PolyA, cloning site: AgeI/HindIII. The target gene was obtained by PCR, ACAA2 (77419-1)-p1: 5′-CTGGGAC AGTGAATACCGGTCGCCACCATGGCCCTGCTACGAGGTGTG-3′, ACA A2 (77419-1)-p2: 5′- GCCTCAGCTATTTAAAGCTTTCATTTGTCGTCATC ATCCTTATAG-3′). The recombinant plasmids were constructed and the products were exchanged into linearized expression vectors, and the primers were identified by PCR, KL77419-p3: CTACTTCAATGAGG AGATGG, KL77419-p4: CATTCTAGTTTTGTGGTTTGTCC. The virus at a dose of $1 × 10^{11}$ genome copies per animal were injected into mice via tail vein after 4-week HFD treatment.

## In vitro cell model of lipid accumulation

Palmitic acid (PA, P0500, Sigma-Aldrich) was dissolved in 50% ethanol to a concentration of 20 mM stock solution. 30% BSA (A8020, Solarbio, China) is stock solution. Then PA solution and BSA solution were mixed together in DMEM to obtain 0.4 mM working solution and 1% BSA working solution. PA (0.4 mM) was added to cultured cells for 24 hours. The cells were then stained with Oil red O (G1262, Solarbio, China) to examine the amount of

lipid accumulation. Intracellular TG and TC levels were measured using a commercially available TG Assay Kit (A110-1-1, Nanjing jiancheng Bioengineering Institute, China) and TC Assay Kit (A111-1-1, Nanjing jiancheng Bioengineering Institute, China) according to the manufacturer's protocol.

## Cell transfection with plasmids or siRNA

Transfection of plasmids or siRNA was carried out with lipofectamine 2000 reagent (Invitrogen, Carlsbad, America) and X-treme gene siRNA transfection reagent (Roche, Basel, Switzerland), respectively. The sequences of siRNAs for human CAND1 were: 5′-GUUAUGAGCU-GUGGAAAUATT-3′, and 5′-AUUUCCACAGCUCAUAACTT-3′. The sequences of siRNAs for mouse CAND1 were: 5′-AACUGGGCAAAU-GUUUCAUTT-3′, and 5′-AUGAAACAUUUGCCCAGUUTT-3′. Full length human CAND1, human androgen receptor plasmids were purchased from Genechem (Shanghai, China).

## Molecular docking of Cullin1, FBXO42 and ACAA2

protein–protein docking was performed to reveal the binding affinity between FBXO42 (AlphaFold prediction), Cullin1 (PDB ID: 1U6G), and ACAA2 (PDB ID: 4C2J) by using the ClusPro2.0 server. ClusPro2.0 provides stiff docking directly using the PIPER tool, a docking software based on fast Fourier transform (FFT) algorithms that undertake exhaustive sampling of the conformational space on a dense grid in order to sample the most near-native structures for more accurate docking structure. The result was shown using the molecular visualization tool PyMOL after receiving the findings from the ClusPro service.

## Real-time quantitative reverse transcriptase-PCR

Total RNA was extracted from cells or tissue using Trizol (Invitrogen, Carlsbad, America) reagent and cDNA was obtained using the Trans-Script All in-one First-strand cDNA Synthesis Supermix for qPCR Kit (TransGen Biotech, Beijing, China). Real-time quantitative PCR was performed in Step One ABI real-time PCR System through SYBR Green Master (Roche, Basel, Switzerland). The primers used were listed in Supplementary Table 3.

## Western bolt

Total protein was extracted by RIPA lysis buffer containing protease inhibitors and was separated by SDS-PAGE and transferred to polyvinylidene difluoride membranes (Pall Corporation, Mexico, USA). The membranes were blocked in StartingBlock (Genscript ProBio, Nanjing, China) and stained with the appropriate primary and secondary antibodies. After washing with PBST (phosphate-buffered saline with Tween), the membranes were scanned and analyzed by ODYSSEY machine (LI-COR, American). The antibodies used are listed in Supplementary Table 4.

## Histological analysis

Liver sections (5 μm) were embedded in paraffin and then stained using hematoxylin and eosin (H&E) to visualize the pattern of lipid accumulation. Lipid droplet accumulation in the liver was visualized using Oil Red O staining of frozen liver sections (8 μm) that were prepared in Tissue-Tek opti-mum cutting temperature (OCT) compound. The histological features of the tissues were observed and imaged using a light microscope (LEICA, Germany).

## Co-Immunoprecipitation (IP) assay

Protein A/G Magnetic Beads (HY-K0202-1) were purchased by MedChemExpress (MCE, New Jersey, American). IP was performed according to the manufacturer's protocol. Briefly, tissues were lysed with ice-cold lysis buffer (21309264, Biosharp, China) and phosphatase inhibitor tablets. Lysates were cleared by centrifugation, and protein was immunoprecipitated with the indicated antibodies and Protein G

Agarose beads at 4 °C overnight. The beads were washed three times with lysis buffer and then heated at 95 °C in loading buffer for 10 min. Then, the beads were removed and the remaining immunocomplexes were collected and subjected to immunoblotting.

## Triglyceride (TG) and Total cholesterol (TC) detection

TG and TC levels were detected by using the TG and TC Kits (Nanjing Jiancheng Bioengineering Institute, Nanjing, China). Liver tissue was cut into small pieces, and PBS was added in EP tube with liver tissue according to the ratio of weight **g**: volume (ml) = 1:9. Liver tissue was mechanically ground on ice and put into the centrifuge at 500 g for 10 minutes. The supernatant was aspirated to get the working solution. 250 μl G or TC assay reagent and 2.5 μl rking solution were added in 96 well plates for 10 minutes at 37 °C. Absorbance was measured at 510 nm and the TG or TC concentrations was calculated according to the Kit formula.

## Insulin detection

Insulin Elisa Kit was purchased from Elabscience (Wuhan, China). Insulin detection was measured according to the Manufacturer's Protocol.

## Luciferase reporter assay

Truncated CAND1 fragments were constructed by LANDM BIOTECH (Guangzhou, China). Luciferase assay kit was purchased from Thermo-Fisher. CAND1, androgen receptor, and rellina luciferase plasmids were transfected into the cells, The luciferase reporter gene fluorescence intensity was measured according to the manufacturer's protocol.

## Mass spectrometry

Shanghai Origingene was commissioned to perform quantitative protein mass spectrometry. L02 hepatocytes were transfected with Si-CAND1 using X-treme gene siRNA transfection reagent. After treatment with PA (0.4 mM) for 24 h, two groups of cell samples were lysed. Then, protein extraction and quantitative mass spectrometry; protein Trypsin digestion; mass spectrometry. Each sample was separated by capillary HPLC using a nanoliter flow rate HPLC liquid phase system. The chromatographic column was equilibrated with 95% liquid A. Each sample was separated by capillary HPLC and analyzed by mass spectrometry using an Orbitrap Fusion mass spectrometer (Thermo Scientific, American).

## GEO database mining

Raw data in the GSE126848 dataset was downloaded from the GEO database (https://www.ncbi.nlm.nih.gov/geo/). Limma 3.50.1 package was used to analyze gene expression levels between different samples, such as Normal vs NAFL, and Normal *vs* NASH. Genes showing |log$_2$ fold change| > 0.5 and adjusted *P* value < 0.05 were considered to present differential expression.

## Statistical analysis

Data are expressed as the mean ± SD of at least three independent experiments for each experimental group. Student's *t* test was used for comparisons between two groups, one-way analysis of variance followed by Tukey corrected post hoc *t* test was used for multi-group comparisons. A value of *P* < 0.05 was considered statistically significant.

## Reporting summary

Further information on research design is available in the Nature Portfolio Reporting Summary linked to this article.

## Data availability

The mass spectrometry proteomics data have been deposited to the ProteomeXchange Consortium via the PRIDE partner repository with the dataset identifier PXD043719. The previously published data sets

re-analysed in this study were obtained from Gene Expression Omnibus (GEO), through the accession code (GSE126848). All other data generated or analyzed in this study are available within the article and its supplementary information files. Source data are provided with this paper.

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

## Acknowledgements

We thank members of the Department of Pharmacology for the discussion and the Department of Hepatopancreatobility for their assistance in our study. This work was supported by the National Natural Science Foundation of China (81870295, Pan).

## Author contributions

X.H., X.Liu, X.Li, and Y.Z. performed experiments, analyzed data, and prepared the manuscript. J.G., Y.Y., Y.J., H.G., C.S., L.Z., J.S., H.B., Z.Z., S.L. and X.Z. helped perform experiments and collect data. L.X., Y.L., and X.Z. oversaw the project and proofread the manuscript. B.Y. and Z.P. designed the project, oversaw the experiments, and prepared the manuscript.

## Competing interests

The authors declare no competing interests.
