## [Peer Review File · Nature Communications]

Cullin-associated and neddylation-dissociated protein 1 (CAND1) alleviates nonalcoholic fatty liver disease by reducing ubiquitinated degradation of acetyl-CoA acyltransferase 2REVIEWER COMMENTS

Reviewer #1 (Remarks to the Author):

In this study, Huang and co-workers found that CAND1 was decreased in the liver of NAFLD patients and high fat diet (HFD)-fed mice. Blockade of CAND1 with genetic method aggravated the development of hepatic steatosis, insulin resistance and inflammation whereas overexpression of CAND1 attenuated these pathological changes. In addition, the authors also proved that CAND1 mitigated NAFLD through inhibiting Cullin1/FBXO42 mediated ACAA2 degradation. And activation of androgen receptor promoted the transcription of CAND1.

This is a substantial body of work; methods are state-of-the-art and identification of each target is plausible. There are several concerns should be addressed.

- 1) What is the cause of the mortality in CAND1 homozygous mice, growth problems or organ dysfunction? Whether the survival curve should be presented.
- 2) Whether the markers of liver fibrosis should be tested.
- 3) Thinking about therapeutic application of the information in the paper and try to translate your findings for clinical practice.
- 4) In methods, the authors used caudal vein injection of ACAA2-AAV8 virus to mice, please note the literature support for the relevant method. And the title "Construction of CAND1 overexpression virus" should be changed to "Construction of ACCA2 overexpression virus".
- 5) For the first time, please use both a full and abbreviated form, as "Skp1, Cul1".
- 6) In Fig 5, there are two bands for the WB of Cullin1, which one is Cullin1?
- 7) The language needs an in-depth revision. Such as in Supplementary Figure 2E, "CAND1" are not labeled correctly.
- 8) In Supplementary Figure 3D and 3E, the annotation is adversed.
- 9) Pay attention to the letter case in this paper. For example, in Supplementary Figure 3E "Cand1" should be "CAND1".

Reviewer #2 (Remarks to the Author):

Article entitled “Cullin-associated and neddylation-dissociated protein 1 (CAND1) alleviates 2 nonalcoholic fatty liver disease by reducing ubiquitinated degradation of acetyl-3 CoA acyltransferase 2” by Huang et al., is very interesting, however, it dampens our interest due to the cell line (LO2 cells) they have used in their study.

Major revision.

1) According to the recent Letter to the Editor by Ralf Weiskirchen claims that : LO2, a misidentified cell line: Some data should be interpreted with caution (Weiskirchen R. Letter to the Editor: LO2, a misidentified cell line: Some data should be interpreted with caution. Hepatology. 2022 Aug 15. doi: 10.1002/hep.32730.)

Dr. Ralf also states that “The cell line LO2, also termed “L-02” or “HL-7702,” was in fact originally introduced as a human fetal hepatocyte cell line. However, 2 years later the cell line was shown to be a derivative of the cervical cancer line HeLa. Unfortunately, this misidentified cell line is still used mistakenly in many laboratories as a hepatocyte derivative. Similar to the “Chang cell line,” this cell line is most frequently used in studies originating from Asia”

Also, this paper proves that LO2 is not hepatocyte cell lines (Ye F, Chen C, Qin J, Liu J, Zheng C. Genetic profiling reveals an alarming rate of cross-contamination among human cell lines used in China. FASEB J. 2015 Oct;29(10):4268-72.)

Here authors used LO2 cell line to prove that CAND1 alleviates 2 nonalcoholic fatty liver disease in Figure 4 and Figure 8. Therefore, authors need to repeat these experiments using proper hepatocyte cell lines like THLE-2 or THLE-3 or HepG2.

2) Authors need to mention the role of acetyl-CoA acyltransferase 2 (ACAA2) in NAFLD in the introduction part.

3) This hormone related study, Therefore, sex of the wild type mice, transgenic mice and knock out mice needs to be mentioned in the manuscript.

4) According to this study Androgen acts as a protective agent for NAFLD. There are studies which show that Androgen receptor antagonist enzalutamide also inhibits Hepatocellular Carcinoma which is an advanced form of NAFLD. How authors answer these contracting roles of Androgen therapy for NAFLD.

5) The methods used especially for animal studies were not properly presented. Eg. In the animal treatment section, the diet and treatment period are not properly specified. It is necessary to reframe the experimental design section and include the specific diet and treatment period details.

6) In the lipid accumulation study section of the in-vitro cell line studies, 30% BSA was used for conjugation with PA.? Conjugation of fatty acids is achieved using just 1-10% BSA. Authors must provide proper reference for this method.

7) In the histopathology section, the authors need to mention the thickness of the liver sections used for staining.

8) Also, authors should specify the magnification at which the histological images were captured.

9) Procedure for extraction for TG and TC in liver tissue should be mentioned in the method part.

10) The author refers to the paper and states that CAND1 knockout contributed to rapid proliferation and high migration of lung cancer cells. Does CAND1 knockout also contribute to NAFLD associated HCC?

11) NAFLD as a Sexual Dimorphic Disease. Does CAND1 play any role in this along with androgens and testosterone?

12) Did you find any cardiac hypertrophy in these CAND1 knockout mice?

13) Authors need to correct some typographic errors eg: mic in line number 254.

Reviewer #3 (Remarks to the Author):

The manuscript entitled "Cullin-1 associated and neddylation-dissociated protein 1 (CAND1) alleviates nonalcoholic fatty liver disease by reducing ubiquitinated degradation of acetyl-3 CoA acyltransferase" by Xiang H., et al. is well written and the efficacy of CAND1 against NAFLD is interest. CAND1 modulates SCF function through its interactions with the CUL1 subunit and functions as an exchange factor for SCF E3 ligases. CAND1 is a cullin-associated protein whose binding is excluded by cullin neddylation. Although early biochemical studies suggested that CAND1 inhibits CRL activity, recently Cand1 has been reported to stimulate assembly of SCF complexes by exchanging the Skp1-F-box protein substrate receptor modules. The authors demonstrated that hepatic CAND1 expression, which is low in murine and human NAFLD and regulated by AR, could prevent NAFLD development via reducing ub-mediated degradation of ACAA protein. Their findings are new and interesting. However, the manuscript has several concerns, and reviewer feels that more information should be required for publication. The detailed comments are attached below.

Major

1. the GEO database of NAFLD patients includes female. Wouldn't it be better to assesses only males for the purpose of the study? Discussion is needed.
2. LO2 cell line was shown to be a derivate of the cervical cancer line HeLa (PMID: 36040018).
3. In Fig 2, 3, no information on body weight, epididymal fat weight and food intake in Wt, cKO, and cKI mice fed high-fat diets for 16 weeks. Not only insulin resistance but also these factors may be affected from dysregulation of various hepatokines due to decreased FAO.
4. In Fig 4B, ACAA2, binds to cullin, was upregulated in the siCAND1 group. However, these LO2 cells were not treated with some proteasome inhibitors. On the other hand, Fig 4C showed that ACCA2 is low in cKO mice. These findings indicates that ubiquitinated ACAA2 was not quickly degraded?
5. In Fig 4E, IP control (IgG lane) is missing.
6. In input immuno-blots of Fig 5, ACAA2 levels look like similar between HFD-fed wt vs cKO, chow-fed Wt vs cKO mice and chow or HFD fed wt mice. The inputs within Fig A-D and Fig

D-E should be the same sample.

7. In Fig 7A, representative gross images of livers of 24-hour fasted wild-type appears quite white though no intrahepatic lipid accumulation on HE-staining section.

8. The role of CAND1 in female cKO mice will be good information for the reader.

9. The levels of testosterone and hepatic AR expression in CAND1-cKO mice should be examined.

10. There could be other pathways for regulating CAND1 expression except for testosterone/AR. This is because CAND1 is also expressed or upregulated in a female mouse liver, AR-negative cell lines (HepG2 and PLC/PR5) and ER α -positive breast cancer. The reporter analysis and the dynamics of AR expression in mouse livers indicate a possible regulation of CAND1 by AR. Not clear that it is a major pathway in the development of NAFLD. Further studies by using AR-deficient mice, orchietomized mice, or AR-negative cell lines are needed.

11. Recently reported, degradation of NRF2 by neddylation of cullin 3 was linked to AGER1 downregulation and promote NASH (PMID: 32657776). The authors should discuss the other targeted molecules by SCF due to depletion of CAND1. For example, in Fig 4B, slc25a24, which binds to cullin is also increased. Slc25a24-knockout mice exhibited has been reported to reduce triglyceride deposition in the liver (PMID: 25599384).

Responses to Reviewer's Comments

We thank the reviewers for their positive comments on our work. The comments are very valuable and constructive, which are very important for us to improve the quality of the work. We have performed a series of additional experiments to strengthen the conclusion of the study.

Reviewer #1:

In this study, Huang and co-workers found that CAND1 was decreased in the liver of NAFLD patients and high fat diet (HFD)-fed mice. Blockade of CAND1 with genetic method aggravated the development of hepatic steatosis, insulin resistance and inflammation whereas overexpression of CAND1 attenuated these pathological changes. In addition, the authors also proved that CAND1 mitigated NAFLD through inhibiting Cullin1/FBXO42 mediated ACAA2 degradation. And activation of androgen receptor promoted the transcription of CAND1.

This is a substantial body of work; methods are state-of-the-art and identification of each target is plausible. There are several concerns should be addressed.

Reply: We would like to express our sincere gratefulness to you for your positive comments on our work, and your suggestions are constructive and very helpful for improving the quality of our manuscript. Please find our point-by-point responses to your individual comments and suggestions as shown below.

1) What is the cause of the mortality in CAND1 homozygous mice, growth problems or organ dysfunction? Whether the survival curve should be presented.

Reply: Thank you for the valuable suggestion. Actually, there were no CAND1 homozygous mice born, which means gametophytic or embryo lethality. We can only use **CAND1 heterozygous mice** for preliminary study. We thus generated hepatocyte-specific knockout mice to accomplish complete deletion of CAND1 in hepatocytes.

2) Whether the markers of liver fibrosis should be tested.

Reply: Thank you for the good suggestion. We used Sirius red staining experiment to specifically identify collagen in fibrotic tissue in liver. The result showed that fibrotic area had no difference between WT and cKO male mice after HFD or NC (**Figure R1, which was shown as Supplementary Figure 6a, b in the manuscript**). Liver fibrosis acts as an end-stage phenotype of NAFLD disease, and we speculate that HFD induction for 16 weeks is not long enough to cause fibrosis in the liver in our study. It has been reported that the same high-fat diet induced liver fibrosis for 38-44 weeks¹.

Figure R1. Examination of liver fibrosis by Sirius red staining. (a) Sirius red staining of liver sections of cKO and WT male mice after 16 weeks of HFD treatment (n=5). (b) Sirius red staining of liver sections of cKI and WT male mice after 16 weeks of HFD treatment (n=5). Scale bar=50 μ m. Magnification 100 \times .

3) Thinking about therapeutic application of the information in the paper and try to translate your findings for clinical practice.

Reply: Thank you for the insightful comment. The data of the study indicated that CAND1 is a potential target for NAFLD. The therapeutic strategy is to upregulate the expression of CAND1, which can be achieved by overexpression of CAND1-carrying adeno-associated virus or nanoparticles with specific liver delivery system, or by the compounds that specifically activate the function or expression of CAND1.

4) In methods, the authors used caudal vein injection of ACAA2-AAV8 virus to mice, please note the literature support for the relevant method. And the title “Construction of CAND1 overexpression virus” should be changed to “Construction of ACCA2 overexpression virus”.

Reply: Thank you for the good suggestion. We added the reference to support the information about caudal vein injection of virus plasmid in Methods (**line 472 Page 23**). We apologized for the writing error. We have corrected the description as “Construction of ACCA2 overexpression virus” in the main text and have once more read through the manuscript and corrected these and some other typos.

5) For the first time, please use both a full and abbreviated form, as “Skp1, Cul1”.

Reply: Thanks for your careful reading. Accordingly, abbreviated “Skp1” (**line 102 Page 6**) for the first time has been replaced by the full name “S-phase kinase associated protein 1”. We finally decided that “Cullin1” was the final expression in the same way as the literature² (**PMID: 35713976**). Abbreviated “Cul1” (**line 324 Page 16**) has been replaced by the full name “Cullin1”.

6) In Fig 5, there are two bands for the WB of Cullin1, which one is Cullin1?

Reply: We apologized for your confusion. According to the Cullin1 antibody instructions (**Proteintech, 12895-1-AP**), the location of Cullin1 is at 90 kDa. The lower band is Cullin1. We have indicated Cullin1 band with a specific arrow (**Figure 5**).

7) The language needs an in-depth revision. Such as in Supplementary Figure 2E, “CAND1” are not labeled correctly.

Reply: Thank you for your patient advice. We have checked the manuscript thoroughly and corrected the typos and language errors. We also invited an English-speaking scientist to help revise the language of our manuscript.

8) In Supplementary Figure 3D and 3E, the annotation is adversed.

Reply: We feel sorry for the mistake. We have corrected the mistake and reviewed all

annotation carefully.

9) Pay attention to the letter case in this paper. For example, in Supplementary Figure 3E “Cand1” should be “CAND1”.

Reply: Thanks for pointing it out. We have corrected it and checked the manuscript thoroughly.

Reviewer #2:

Article entitled “Cullin-associated and neddylation-dissociated protein 1 (CAND1) alleviates 2 nonalcoholic fatty liver disease by reducing ubiquitinated degradation of acetyl-3 CoA acyltransferase 2” by Huang et al., is very interesting, however, it dampens our interest due to the cell line (LO2 cells) they have used in their study.

Reply: We are grateful to you for your very positive comments on our work and constructive suggestions for improving our manuscript. We have performed several sets of additional experiments to address your concerns. We repeated all the *in vitro* experiments with AML12, THLE-2 and HepG2 cells.

1) According to the recent Letter to the Editor by Ralf Weiskirchen claims that : LO2, a misidentified cell line: Some data should be interpreted with caution (Weiskirchen R. Letter to the Editor: LO2, a misidentified cell line: Some data should be interpreted with caution. Hepatology. 2022 Aug 15. doi: 10.1002/hep.32730.)

Dr. Ralf also states that “The cell line LO2, also termed “L-02” or “HL-7702,” was in fact originally introduced as a human fetal hepatocyte cell line. However, 2 years later the cell line was shown to be a derivative of the cervical cancer line HeLa. Unfortunately, this misidentified cell line is still used mistakenly in many laboratories as a hepatocyte derivative. Similar to the “Chang cell line,” this cell line is most frequently used in studies originating from Asia”

Also, this paper proves that LO2 is not hepatocyte cell lines (Ye F, Chen C, Qin J, Liu J, Zheng C. Genetic profiling reveals an alarming rate of cross-contamination among human cell lines used in China. FASEB J. 2015 Oct;29(10):4268-72.)

Here authors used LO2 cell line to prove that CAND1 alleviates 2 nonalcoholic fatty liver disease in Figure 4 and Figure 8. Therefore, authors need to repeat these experiments using proper hepatocyte cell lines like THLE-2 or THLE-3 or HepG2.

Reply: Thank you for your kindly remind and professional comments. We have critically considered this issue and repeated LO2 cell-associated experiments using AML12, THLE-2 and HepG2 cells and obtained the similar results (**Figure 1, Figure**

4, Figure 8, Supplementary Figure 2, Supplementary Figure 8).

2) Authors need to mention the role of acetyl-CoA acyltransferase 2 (ACAA2) in NAFLD in the introduction part.

Reply: Thank you for the valuable suggestion. We have added information of ACAA2 in NAFLD in the Introduction part (**line 94-99 Page 6**).

3) This hormone related study, Therefore, sex of the wild type mice, transgenic mice and knock out mice needs to be mentioned in the manuscript.

Reply: We appreciate the constructive comments. Wild type mice, CAND1 heterozygous, cKO mice and cKI mice are male. We clarified sex of mice in the manuscript.

4) According to this study Androgen acts as a protective agent for NAFLD. There are studies which show that Androgen receptor antagonist enzalutamide also inhibits Hepatocellular Carcinoma which is an advanced form of NAFLD. How authors answer these contracting roles of Androgen therapy for NAFLD.

Reply: Thank you for the insightful comment. In our study, we demonstrated a protective role of Androgen/AR signaling pathway in NAFLD. Enzalutamide, as a second generation of AR antagonist, has been reported that co-targeting AR (Enzalutamide) and mTOR shows significant synergistic anti-Hepatocellular Carcinoma (HCC) activity and decreases tumor burden by inducing apoptosis in vivo³. Moreover, early clinical trials on anti-androgen and bicalutamide (first generation of AR antagonist) therapies in liver cancer met with disappointing results, producing no apparent clinical benefits^{4,5}. We think one possible explanation is the different pathological property of NAFLD and HCC.

Furthermore, although there is sound evidence that androgen/AR promote hepatocarcinogenesis and HCC development in the early stages, there is less evidence showing a direct link between androgen/AR signaling and HCC progression in advanced stages⁶. Several articles have reported that AR might suppress HCC

metastasis in the advanced stage of HCC^{7,8}. A clear understanding of the mechanisms governing these opposite and dual functions of AR in HCC progression might lead to the development of different therapeutic approaches for HCC at different stages. Androgen may also play a different role at different stages of NAFLD, which highlights the importance of the precise therapeutic window.

5) The methods used especially for animal studies were not properly presented. Eg. In the animal treatment section, the diet and treatment period are not properly specified. It is necessary to reframe the experimental design section and include the specific diet and treatment period details.

Reply: Thank you for the valuable comment. We have revised the animal experiment details such as the specific diet and treatment period details in methods part (**line 422-424 Page 21**).

6) In the lipid accumulation study section of the in-vitro cell line studies, 30% BSA was used for conjugation with PA.? Conjugation of fatty acids is achieved using just 1-10% BSA. Authors must provide proper reference for this method.

Reply: We apologized for your confusion because of the inaccuracy we described. 30% BSA was stock solution. Final working solution concentration was 1% BSA, which was used to conjugate fatty acids. We have clarified the description (**line 487-489 Page 24**).

7) In the histopathology section, the authors need to mention the thickness of the liver sections used for staining.

Reply: Thank you for the good suggestion. The sections were 5 µm thick for H&E staining and 8 µm thick for Oil Red O staining. We have added this information in the Methods part (**line 530 and 533 Page 26**).

8) Also, authors should specify the magnification at which the histological images were captured.

Reply: Thank you for the helpful suggestion. We have specified the magnification of

the histological images in the Figure legends (**Figure 2g, h, Figure 3g, h, Figure 6d, e, Figure 7a, b, Figure 8h, Supplementary Figure 2b, f, j, n, r, v, Supplementary Figure 3e, Supplementary Figure 6a, b, Supplementary Figure 7f**).

9) Procedure for extraction for TG and TC in liver tissue should be mentioned in the method part.

Reply: Thank you for the useful suggestion. We have added this information in Methods part (**line 548-553 Page 27**).

10) The author refers to the paper and states that CAND1 knockout contributed to rapid proliferation and high migration of lung cancer cells. Does CAND1 knockout also contribute to NAFLD associated HCC?

Reply: Thank you for the constructive comment. Numerous hepatic nodules characterized by hepatic hyperplasia, intrahepatic cysts and liver fibrosis are common pathological features of HCC⁹. Hepatic hyperplasia nodules, intrahepatic cysts and liver fibrosis were not shown in WT-HFD and cKO-HFD male mice (**Figure 2g, Figure R2a**). The expression of α FP (Alpha-Fetoprotein) is directly associated with hepatocyte differentiation and HCC progression¹⁰. More importantly, α FP is the most commonly used noninvasive marker for HCC diagnosis¹¹. α FP Elisa result showed that there was no statistical difference in WT-HFD and cKO-HFD male mice. These results suggest that CAND1 does not contribute to NAFLD-HCC progression at least at the time point employed in this study.

Figure R2. (a) Sirius red staining of liver sections of cKO and WT male mice after 16 weeks of HFD treatment (n=5). (b) αFP levels in cKO and WT male mice after 16 weeks of HFD treatment (n=8). * $P < 0.05$ vs WT-HFD group, unpaired two-sided Student's *t*-test. The data were shown as means \pm SD of independent biological replicates. ns, not significant difference. Scale bar=50 μ m. Magnification 100 \times .

11) NAFLD as a Sexual Dimorphic Disease. Does CAND1 play any role in this along with androgens and testosterone?

Reply: Thank you for the insightful comment. To answer the question, we performed experiments in cKO female mice after HFD. Consistent with the results obtained from cKO male mice, WT-HFD and cKO-HFD female mice had comparable food intake and body weight (**Figure R3a, b, which was shown as Supplementary Fig. 7a, b in the manuscript**). The LB/BW ratio was increased in cKO-HFD female mice (**Figure R3c, which was shown as Supplementary Fig. 7c in the manuscript**). Moreover, lipid accumulation was more severe in the livers of cKO-HFD than WT-HFD female mice (**Figure R3d-f, which was shown as Supplementary Fig. 7d-f in the manuscript**). These results indicate that CAND1 deficiency in hepatocytes also aggravated HFD-induced hepatic steatosis in female mice. CAND1 does not show any sexual dimorphic effects.

Figure R3. Hepatocyte-specific CAND1 deficiency exacerbates HFD-induced hepatic steatosis in female mice. (a) Food intake of CAND1 cKO mice and WT female mice after 12 weeks of NC or HFD feeding (n=7). (b) Body weight in indicated groups (n=7). (c) LW/BW ratio (n=9-10). (d, e) TG and TC content in indicated groups (n=6). (g) H&E and Oil red O staining of liver sections (n=6). Scale bar=50 μm. Magnification 200×. * $P < 0.05$ vs WT-HFD (female) group, unpaired two-sided Student's t -test. The data were shown as means \pm SD of independent biological replicates. LW/BW, liver weight/body weight. TG, Triglyceride. TC, Total cholesterol.

12) Did you find any cardiac hypertrophy in these CAND1 knockout mice?

Reply: Thank you for the good suggestion. Our group has previously reported that CAND1 knockout mice did not develop cardiac hypertrophy spontaneously, while they significantly developed cardiac hypertrophy and heart failure in response to transaortic constriction¹².

13) Authors need to correct some typographic errors eg: mic in line number 254.

Reply: Thanks for pointing it out. We have corrected it and checked the manuscript thoroughly.

Reviewer #3:

The manuscript entitled “Cullin-1 associated and neddylation-dissociated protein 1 (CAND1) alleviates nonalcoholic fatty liver disease by reducing ubiquitinated degradation of acetyl-3 CoA acyltransferase” by Xiang H., et al. is well written and the efficacy of CAND1 against NAFLD is interest. CAND1 modulates SCF function through its interactions with the CUL1 subunit and functions as an exchange factor for SCF E3 ligases. CAND1 is a cullin-associated protein whose binding is excluded by cullin neddylation. Although early biochemical studies suggested that CAND1 inhibits CRL activity, recently Cand1 has been reported to stimulate assembly of SCF complexes by exchanging the Skp1-F-box protein substrate receptor modules. The authors demonstrated that hepatic CAND1 expression, which is low in murine and human NAFLD and regulated by AR, could prevent NAFLD development via reducing ub-mediated degradation of ACAA protein. Their findings are new and interesting. However, the manuscript has several concerns, and reviewer feels that more information should be required for publication. The detailed comments are attached below.

Reply: Thank you for the comments concerning our manuscript. Those comments are all valuable and very helpful for revising and improving our paper, as well as the important guiding significance to our research. We have studied the comments carefully and have performed additional experiments to address these issues.

1. the GEO database of NAFLD patients includes female. Wouldn't it be better to assesses only males for the purpose of the study? Discussion is needed.

Reply: Thanks for the constructive suggestion. Accordingly, we re-analyzed male NAFLD patients of GEO database and CAND1 was also one of the differentially expressed genes (**Figure R4, which was shown as Supplementary Figure 1 in the manuscript**). We also discussed this issue in the Discussion part.

Figure R4. Deregulated genes in male patients with NAFLD. (a) Venn diagram highlighting significantly upregulated and downregulated genes in NAFL and NASH patients compared to Normal volunteers. (b) Enriched GO term process associated with protein ubiquitination from gene enrichment analysis on overlapping genes. (c) Heat map depicting relative expression of genes involved with protein ubiquitination.

2. LO2 cell line was shown to be a derivate of the cervical cancer line HeLa (PMID:

36040018).

Reply: Thank you for your kindly remind and professional comments. We repeated L02 cell-associated experiments using AML12, THLE-2 and HepG2 cells and obtained the similar results (**Figure 1, Figure 4, Figure 8, supplementary Figure 2, supplementary Figure 8**).

3. In Fig 2, 3, no information on body weight, epididymal fat weight and food intake in Wt, cKO, and cKI mice fed high-fat diets for 16 weeks. Not only insulin resistance but also these factors may be affected from dysregulation of various hepatokines due to decreased FAO.

Reply: Thank you for the helpful suggestion. Accordingly, we measured body weight, epididymal fat weight and food intake in WT, cKO and cKI male mice after HFD or NC. Of note, there was no statistical difference in food intake between WT and cKO male mice after HFD or NC treatment (**Figure R5a, which was shown as Figure 2a in the manuscript**). The body weight and epididymal fat weight of WT and cKO male mice were higher after 16 weeks HFD than NC, but WT-HFD and cKO-HFD male mice had comparable body weight and epididymal fat weight (**Figure R5b, c, which was shown as Figure 2b, c in the manuscript**). Similar to cKO results, there was no statistical difference in food intake between WT and cKI male mice after HFD or NC treatment (**Figure R5d, which was shown as Figure 3a in the manuscript**). The body weight and epididymal fat weight of WT and cKI male mice were higher after 16 weeks HFD than NC, but WT-HFD and cKI-HFD male mice had comparable body weight and epididymal fat weight (**Figure R5e, f, which was shown as Figure 3b, c in the manuscript**).

Figure R5. Body weight, epididymal fat weight and food intake in WT, cKO and cKI male mice after HFD or NC. (a) Food intake of CAND1 cKO and WT male mice after 16 weeks of NC or HFD feeding (n=7). (b) Body weight in indicated groups (n=9-10). (c) Epididymal fat weight (n=8). (d) Food intake of cKI and WT male mice after 16 weeks of NC or HFD feeding (n=7). (e) Body weight in indicated groups (n=9-10). (f) Epididymal fat weight (n=8). * $P < 0.05$ vs WT group, one-way ANOVA with Tukey's multiple comparisons test. The data were shown as means \pm SD of independent biological replicates.

4. In Fig 4B, ACAA2, binds to cullin, was upregulated in the siCAND1 group. However, these LO2 cells were not treated with some proteasome inhibitors. On the other hand, Fig 4C showed that ACCA2 is low in cKO mice. These findings indicates that ubiquitinated ACAA2 was not quickly degraded?

Reply: Thank you for the constructive suggestion. CAND1 is a protein exchange factor that accelerates the rate at which Cullin1-Rbx1 equilibrates with multiple F-box Protein-Skp1 modules. Depletion of CAND1 from cells impedes recruitment of new F-box proteins to preexisting Cullin1 and profoundly alters the cellular landscape of SCF complexes, which lead to slow degradation of substrate proteins¹³. To address this

issue, we measured the degradation rate of ACAA2 by inhibiting its ubiquitinated degradation using MG132 in THLE-2 cells. MG132 treatment for 6 hours did not change ACAA2 protein level in PA-induced THLE-2 cells with CAND1 knockdown (**Figure R6**). Treatment with MG132 for 12 or 24 hours increased the protein expression of ACAA2 (**Figure R6**). These results indicated that ubiquitinated ACAA2 is not quickly degraded with 6 hours in PA-induced THLE-2 cells after CAND1 knockdown. The estimated half-life for ACAA2 was 30 hours in mammalian reticulocytes by “ProtParam”, which also indicated that the degradation rate of ACAA2 protein was slow.

Figure R6. ACAA2 protein levels in PA-induced THLE-2 cells after knockdown of CAND1 treated with MG132 or NC in indicated time (n=3). * $P < 0.05$, unpaired two-sided Student’s *t*-test. The data were shown as means \pm SD of independent biological replicates. ns, not significant difference.

5. In Fig 4E, IP control (IgG lane) is missing.

Reply: Thank you for reminding. We have added IgG as IP negative control (**Figure R7a, b, which was shown as Figure 4g, h in the manuscript**).

Figure R7. CAND1 controls the ubiquitination and degradation of fatty acid β -oxidation gene ACAA2. (a) Ubiquitination of ACAA2 in the livers of WT and cKO mice after 16 weeks of NC or HFD feeding (n=3). (b) Ubiquitination of ACAA2 in the livers of WT and cKI mice after 16 weeks of NC or HFD feeding (n=3). * $P < 0.05$ vs WT group, # $P < 0.05$ vs WT-HFD group, one-way ANOVA with Tukey's multiple comparisons test. The data were shown as means \pm SD of independent biological replicates.

6. In input immuno-blots of Fig 5, ACAA2 levels look like similar between HFD-fed wt vs cKO, chow-fed Wt vs cKO mice and chow or HFD fed wt mice. The inputs within Fig A-D and Fig D-E should be the same sample.

Reply: Thank you for the good comment. We repeated the immuno-blot experiments with the same samples for input. Now the quality of inputs is better, and we can easily see the difference among groups. We used one single input for Fig A-C and Fig D-E, respectively, as your kind recommendation. The inputs within Fig 5A-C and Fig 5D-E were the same sample, and we have rearranged **Figure 5 (Figure 5a, b)**.

7. In Fig 7A, representative gross images of livers of 24-hour fasted wild-type

appears quite white though no intrahepatic lipid accumulation on HE-staining section.

Reply: Thank you for the helpful suggestion. We have replaced previous HE images and added new experimental data of Oil red O staining accordingly, which is more indicative of intrahepatic lipid accumulation in fasted cKO male mice compared to fasted WT male mice (**Figure R8, which was shown as Figure 7b in the manuscript**).

Figure R8. Hepatocyte-specific CAND1 knockout promotes starvation-induced fatty liver in male mice. Oli red O and H&E staining from WT and cKO male mice that were fasted for 24h (n=5). Scale bar=50 μ m. Oil red O magnification 200 \times , H&E magnification 400 \times .

8. The role of CAND1 in female cKO mice will be good information for the reader.

Reply: Thank you for the insightful suggestion. To answer the question, we performed experiments in cKO female mice after HFD. Consistent with the results in cKO male mice, WT-HFD and cKO-HFD female mice had comparable food intake and body weight (**Figure R9a, b, which was shown as Supplementary Fig. 7a, b in the manuscript**). The LB/BW ratio was increased in cKO-HFD female mice (**Figure R9c, which was shown as Supplementary Fig. 7c in the manuscript**). Moreover, lipid

accumulation was more severe in the livers of cKO-HFD than WT-HFD female mice (**Figure R9d-f, which was shown as Supplementary Fig. 7d-f in the manuscript**). These results indicate that CAND1 deficiency in hepatocytes also aggravates HFD-induced hepatic steatosis in female mice.

Figure R9. Hepatocyte-specific CAND1 deficiency exacerbates HFD-induced hepatic steatosis in female mice. (a) Food intake of CAND1 cKO mice and WT female mice after 12 weeks of NC or HFD feeding (n=7). (b) Body weight in indicated groups (n=7). (c) LW/BW ratio (n=9-10). (d, e) TG and TC content in indicated groups (n=6). (g) H&E and Oil red O staining of liver sections (n=6). Scale bar=50 μm. Magnification 200×. * $P < 0.05$ vs WT-HFD (female) group, unpaired two-sided Student's *t*-test. The data were shown as means ± SD of independent biological replicates. LW/BW, liver weight/body weight. TG, Triglyceride. TC, Total cholesterol.

9. The levels of testosterone and hepatic AR expression in CAND1-cKO mice should be examined.

Reply: Thank you for the insightful suggestion. We detected serum testosterone levels

and liver AR expression in cKO male mice, and found that cKO male mice had lower AR expression and lower serum testosterone levels than WT male mice (**Figure R10a, b**, which was shown as **Supplementary Figure 10a, b in the manuscript**). This provides additional evidence to support reciprocal feedback regulation relationship between CAND1 and AR in male mice.

Figure R10. The levels of testosterone and hepatic AR expression in CAND1-cKO mice. (a) mRNA level of AR of WT and cKO mice (n=6). (b) Plasma testosterone levels of WT and cKO mice (n=6). * $P < 0.05$ vs WT mice, unpaired two-sided Student's *t*-test.

10. There could be other pathways for regulating CAND1 expression except for testosterone/AR. This is because CAND1 is also expressed or upregulated in a female mouse liver, AR-negative cell lines (HepG2 and PLC/PR5) and ER α -positive breast cancer. The reporter analysis and the dynamics of AR expression in mouse livers indicate a possible regulation of CAND1 by AR. Not clear that it is a major pathway in the development of NAFLD. Further studies by using AR-deficient mice, orchietomized mice, or AR-negative cell lines are needed.

Reply: Thank you for the useful suggestion. Studies have shown that androgen/AR signaling suppresses the development of steatosis in NAFLD. Androgens were proven to be suppressors of the disease in rats fed with a high-fat diet¹⁴. Furthermore, C57BL/6 mice receiving the anti-androgen hydroxyflutamide were shown to have a higher incidence of NAFLD¹⁵. In an HFD-induced NAFLD mouse model, the loss of AR in the whole body (GARKO) was shown to lead to higher insulin insensitivity and the

development of diabetes¹⁶. Similar results were confirmed in mice that only lacked hepatic AR (L-ARKO), indicating that hepatic AR might play negative roles in HFD-induced NAFLD¹⁷. Furthermore, aged male mice lacking hepatic AR developed hepatic micro-vesicle steatosis, whereas WT aged mice did not¹⁷. In our study, we demonstrated that AR acted as a transcription factor for CAND1 and regulated CAND1 expression in the development NAFLD. It has been reported that testosterone protected HFD diet-induced NAFLD in castrated male rats by reducing hepatic macrovesicular fat accumulation and inflammation¹⁸.

To further highlight the importance of AR/CAND1 pathway in NAFLD, we carried out a series of experiments in AR-positive (AML12) and AR-negative (THLE-2 and HepG2) cells. We found that PA-induced lipid accumulation was similar in siCAND1 and siAR group (**Figure R11a-c, which was shown as Figure 8h-j int manuscript**). Concomitant knockdown of AR and CAND1 exacerbated PA-induced lipid accumulation compared with knockdown of CAND1 group in AML12 cells (**Figure R11a-c, which was shown as Figure 8h-j int manuscript**), while overexpression of CAND1 attenuated AR deficiency-caused lipid accumulation in AML12 cells after PA treatment (**Figure R11a-c, which was shown as Figure 8h-j int manuscript**). These results firmly confirmed the important role of AR-CAND1 axis in NAFLD development.

Interestingly, we found that knockdown of CAND1 aggravated lipid accumulation after PA induction in AR-negative (THLE-2 and HepG2) cells (**Figure R12, which was shown as Supplementary Figure 2 in the manuscript**). Conversely, overexpression of CAND1 mitigated lipid accumulation after PA induction in AR-negative (THLE-2 and HepG2) cells (**Figure R12, which was shown as Supplementary Figure 2 in the manuscript**). These data demonstrated that CAND1 promoted lipid metabolism and reduced fat accumulation in AR-negative cells.

Given the regulatory effect of CAND1 on lipid accumulation in female mice and AR-negative cell lines (THLE-2 and HepG2), there may be other pathways involved in regulating CAND1 expression and the development of NAFLD in female mice.

Figure R11. AR-CAND1 axis participates in hepatocyte lipid metabolism. (a) Oil red O staining of SI-CAND1, SI-AR, SI-CAND1+SI-AR and SI-AR+OE-CAND1 groups after PA treatment (n=6). (b, c) TG and TC contents in the indicated group after PA treatment (n=6, * $P < 0.05$, one-way ANOVA with Tukey's multiple comparisons test). The data were shown as means \pm SD of independent biological replicates. ns, not significant difference. TG, Triglyceride. TC, Total cholesterol.

Figure R12. CAND1 alleviates PA-induced lipid accumulation in cultured THLE-2 and HepG2 cells. (a) CAND1 knockdown efficiency by Western blotting (n=4). (b) Oil red O staining of PA treated THLE-2 cells after knockdown of CAND1 (n=6). (c, d) Knockdown of CAND1 on TG and TC content of PA treated THLE-2 cells (n=6). (e) Western blotting of CAND1 in THLE-2 cells transfected with CAND1 overexpressing plasmids (n=4). (f) Oil red O staining of PA treated THLE-2 cells after CAND1 overexpression (n=6). (g, h) TG and TC content of PA treated THLE-2 cells after CAND1 overexpression (n=6). (i) CAND1 knockdown efficiency by Western blotting (n=4). (j) Oil red O staining of PA treated HepG2 cells after knockdown of CAND1

(n=6). (k, l) Knockdown of CAND1 on TG and TC content of PA treated HepG2 cells (n=6). (m) Western blotting of CAND1 in HepG2 cells transfected with CAND1 overexpressing plasmids (n=4). (n) Oil red O staining of PA treated HepG2 cells after CAND1 overexpression (n=6). (o, p) TG and TC content of PA treated HepG2 cells after CAND1 overexpression (n=6). * $P < 0.05$ vs NC group or NC+PA group, unpaired two-sided Student's *t*-test or one-way ANOVA. The data were shown as means \pm SD of independent biological replicates. TG, Triglyceride; TC, Total cholesterol; NC, negative control. Scale bar=50 μ m. Magnification 200 \times

11 . Recently reported, degradation of NRF2 by neddylation of cullin 3 was linked to AGER1 downregulation and promote NASH (PMID: 32657776). The authors should discuss the other targeted molecules by SCF due to depletion of CAND1. For example, in Fig 4B, slc25a24, which binds to cullin is also increased. Slc25a24-knockout mice exhibited has been reported to reduce triglyceride deposition in the liver (PMID: 25599384).

Reply: Thank you for the constructive comment. Accordingly, we discussed these findings in the discussion section (**line 367-379 Page 18 and 19**). “However, considering the fact that CAND1 can regulate the expression of a large number of substrate proteins, we cannot rule out the involvement of other target substrates in the regulation of CAND1 on NAFLD. Solute Carrier Family 25 Member 24 (SLC25A24) is one of the substrates of CAND1 we found in the study. Deletion of SLC25A24 in female mice led to lower body weights, white adipose tissue weights and liver weight when consuming the high-fat diet¹⁹. It is possible that SLC25A24 may regulate the effects of CAND1 on NAFLD. Furthermore, increased neddylation of CUL3, an essential regulatory element of the NRF2 complex, by AGEs rendered NRF2 less stable and destined for degradation, which led to advanced glycation end-products clearance receptor (AGER1) downregulation and promoted NASH²⁰. This cascade may be disturbed due to loss of CAND1 in early NAFLD. Therefore, lack of deep study on multiple downstream targets of CAND1 is a limitation of the current study.”

References:

- 1 Hansen, H. H. *et al.* Human translatability of the GAN diet-induced obese mouse model of non-alcoholic steatohepatitis. *BMC gastroenterology* **20**, 210, doi:10.1186/s12876-020-01356-2 (2020).
- 2 Yuan, T. *et al.* SDHA/B reduction promotes hepatocellular carcinoma by facilitating the deNEDDylation of cullin1 and stabilizing YAP/TAZ. *Hepatology (Baltimore, Md.)*, doi:10.1002/hep.32621 (2022).
- 3 Zhang, H. *et al.* Significance and mechanism of androgen receptor overexpression and androgen receptor/mechanistic target of rapamycin cross-talk in hepatocellular carcinoma. *Hepatology (Baltimore, Md.)* **67**, 2271-2286, doi:10.1002/hep.29715 (2018).
- 4 Verslype, C. *et al.* The management of hepatocellular carcinoma. Current expert opinion and recommendations derived from the 10th World Congress on Gastrointestinal Cancer, Barcelona, 2008. *Annals of oncology : official journal of the European Society for Medical Oncology* **20 Suppl 7**, viii1-viii6, doi:10.1093/annonc/mdp281 (2009).
- 5 Llovet, J. M. *et al.* Sorafenib in advanced hepatocellular carcinoma. *The New England journal of medicine* **359**, 378-390, doi:10.1056/NEJMoa0708857 (2008).
- 6 Ma, W. L., Lai, H. C., Yeh, S., Cai, X. & Chang, C. Androgen receptor roles in hepatocellular carcinoma, fatty liver, cirrhosis and hepatitis. *Endocrine-related cancer* **21**, R165-182, doi:10.1530/erc-13-0283 (2014).
- 7 Ma, W. L. *et al.* Hepatic androgen receptor suppresses hepatocellular carcinoma metastasis through modulation of cell migration and anoikis. *Hepatology (Baltimore, Md.)* **56**, 176-185, doi:10.1002/hep.25644 (2012).
- 8 Tavian, D. *et al.* Androgen receptor mRNA under-expression in poorly differentiated human hepatocellular carcinoma. *Histology and histopathology* **17**, 1113-1119, doi:10.14670/hh-17.1113 (2002).
- 9 Ambade, A., Satishchandran, A., Gyongyosi, B., Lowe, P. & Szabo, G. Adult mouse model of early hepatocellular carcinoma promoted by alcoholic liver disease. *World journal of gastroenterology* **22**, 4091-4108, doi:10.3748/wjg.v22.i16.4091 (2016).
- 10 Abelev, G. I. & Eraisier, T. L. Cellular aspects of alpha-fetoprotein reexpression in tumors. *Seminars in cancer biology* **9**, 95-107, doi:10.1006/scbi.1998.0084 (1999).
- 11 Drinane, M. C. & Shah, V. H. Alcoholic hepatitis: Diagnosis and prognosis. *Clinical liver disease* **2**, 80-83, doi:10.1002/cld.164 (2013).
- 12 Li, X. *et al.* Cullin-associated and neddylation-dissociated 1 protein (CAND1) governs cardiac hypertrophy and heart failure partially through regulating calcineurin degradation. *Pharmacological research* **182**, 106284, doi:10.1016/j.phrs.2022.106284 (2022).
- 13 Pierce, N. W. *et al.* Cand1 promotes assembly of new SCF complexes through

- dynamic exchange of F box proteins. *Cell* **153**, 206-215, doi:10.1016/j.cell.2013.02.024 (2013).
- 14 Zhang, H. *et al.* Differential effects of estrogen/androgen on the prevention of nonalcoholic fatty liver disease in the male rat. *Journal of lipid research* **54**, 345-357, doi:10.1194/jlr.M028969 (2013).
- 15 Takahashi, Y., Soejima, Y. & Fukusato, T. Animal models of nonalcoholic fatty liver disease/nonalcoholic steatohepatitis. *World journal of gastroenterology* **18**, 2300-2308, doi:10.3748/wjg.v18.i19.2300 (2012).
- 16 Lin, H. Y. *et al.* Insulin and leptin resistance with hyperleptinemia in mice lacking androgen receptor. *Diabetes* **54**, 1717-1725, doi:10.2337/diabetes.54.6.1717 (2005).
- 17 Lin, H. Y. *et al.* Increased hepatic steatosis and insulin resistance in mice lacking hepatic androgen receptor. *Hepatology (Baltimore, Md.)* **47**, 1924-1935, doi:10.1002/hep.22252 (2008).
- 18 Jia, Y. *et al.* Testosterone protects high-fat/low-carbohydrate diet-induced nonalcoholic fatty liver disease in castrated male rats mainly via modulating endoplasmic reticulum stress. *American journal of physiology. Endocrinology and metabolism* **314**, E366-e376, doi:10.1152/ajpendo.00124.2017 (2018).
- 19 Urano, T., Shiraki, M., Sasaki, N., Ouchi, Y. & Inoue, S. SLC25A24 as a novel susceptibility gene for low fat mass in humans and mice. *The Journal of clinical endocrinology and metabolism* **100**, E655-663, doi:10.1210/jc.2014-2829 (2015).
- 20 Dehnad, A. *et al.* AGER1 downregulation associates with fibrosis in nonalcoholic steatohepatitis and type 2 diabetes. *The Journal of clinical investigation* **130**, 4320-4330, doi:10.1172/jci133051 (2020).

REVIEWERS' COMMENTS

Reviewer #1 (Remarks to the Author):

The authors have thoroughly revised the manuscript according to the reviewers' comments. In my opinion, the changes have made the paper more interesting and clear. I have no future comments.

Reviewer #2 (Remarks to the Author):

This is revised version of the manuscript(R1). The authors have answered all the comments by the reviewers. Therefore, I am suggesting to accept the manuscript.

Reviewer #3 (Remarks to the Author):

The Authors have addressed all of my concerns with the original manuscript.